# BCDiff: Bidirectional Consistent Diffusion for Instantaneous Trajectory Prediction

**Rongqing Li**
Beijing Institute of Technology
lirongqing99@gmail.com

**Changsheng Li** [*]
Beijing Institute of Technology
lcs@bit.edu.cn

**Dongchun Ren**
ALLRIDE.AI
Dongchun.ren@allride.ai

**Guangyi Chen**
CMU & MBZUAI
guangyichen1994@gmail.com

**Ye Yuan**
Beijing Institute of Technology
yuan-ye@bit.edu.cn

**Guoren Wang**
Beijing Institute of Technology
wanggrbit@126.com

## Abstract

The objective of pedestrian trajectory prediction is to estimate the future paths of pedestrians by leveraging historical observations, which plays a vital role in ensuring the safety of self-driving vehicles and navigation robots. Previous works usually rely on a sufficient amount of observation time to accurately predict future trajectories. However, there are many real-world situations where the model lacks sufficient time to observe, such as when pedestrians abruptly emerge from blind spots, resulting in inaccurate predictions and even safety risks. Therefore, it is necessary to perform trajectory prediction based on instantaneous observations, which has rarely been studied before. In this paper, we propose a **B**i-directional **C**onsistent **Diff**usion framework tailored for instantaneous trajectory prediction, named **BCDiff**. At its heart, we develop two coupled diffusion models by designing a mutual guidance mechanism which can bidirectionally and consistently generate unobserved historical trajectories and future trajectories step-by-step, to utilize the complementary information between them. Specifically, at each step, the predicted unobserved historical trajectories and limited observed trajectories guide one diffusion model to generate future trajectories, while the predicted future trajectories and observed trajectories guide the other diffusion model to predict unobserved historical trajectories. Given the presence of relatively high noise in the generated trajectories during the initial steps, we introduce a gating mechanism to learn the weights between the predicted trajectories and the limited observed trajectories for automatically balancing their contributions. By means of this iterative and mutually guided generation process, both the future and unobserved historical trajectories undergo continuous refinement, ultimately leading to accurate predictions. Essentially, BCDiff is an encoder-free framework that can be compatible with existing trajectory prediction models in principle. Experiments show that our proposed BCDiff significantly improves the accuracy of instantaneous trajectory prediction on the ETH/UCY and Stanford Drone datasets, compared to related approaches.

---

[*]Changsheng Li (lcs@bit.edu.cn) is the corresponding author

37th Conference on Neural Information Processing Systems (NeurIPS 2023).

# 1 Introduction

Pedestrian trajectory prediction aims to predict future trajectories conditioned on their past movements, which is an important task for autonomous driving [25, 57] and navigation robot [4]. Previous pedestrian trajectory prediction approaches usually rely on long enough observation time (typically, 2 to 3 seconds) for a pedestrian to precisely predict the future trajectories [53, 48, 54, 13]. However, in many real-world situations, e.g., when pedestrians suddenly emerge from blind spots and are in close proximity to autonomous vehicles, traditional trajectory prediction methods do not have ample time to collect a sufficient number of locations. This leads to sub-optimal prediction performance and potentially unsafe behaviors in the decision-making of autonomous vehicles and robots. Therefore, it is quite necessary to forecast future trajectories based on limited or instantaneous observations.

The prediction of instantaneous trajectories for pedestrians is a highly challenging task due to the limited observation time. In some cases, as extreme as it can be, merely two frames of locations can be observed. In the face of such a challenging task, there have been only a few academic works to date. MOE [46] is the first to propose the problem of instantaneous trajectory prediction, and thus is the most relevant to ours. MOE incorporates scene context information into limited observations and introduces the masked trajectory complement and context restoration as self-supervised tasks to pretrain the model. However, since MOE only utilizes instantaneous temporal information acquired from limited trajectories, it might be hard to accurately predict the future trajectories of a pedestrian with complex behavior such as turning and yielding. DTO [34] focuses on lowering the influence of noise introduced by incorrect detection and tracking, and attempts to employ limited observed trajectory to alleviate this problem. It utilizes the knowledge distillation technique to distill knowledge from a teacher model trained with an ample amount of long observations, and transfer the knowledge to a student model receiving fewer observations as input. Although these approaches have shown some effectiveness in instantaneous trajectory prediction, the representation of a pedestrian is restricted to two frames of locations, which contains extremely limited temporal information.

In this paper, we propose BCDiff, a bidirectional consistent diffusion framework specifically designed for the instantaneous trajectory prediction task. As we know, the diffusion model is a generative model, which has been successfully applied to various generation tasks, including image synthesis [39, 33], image denoising [23], etc. Different from them, we leverage the diffusion model for instantaneous trajectory prediction. We devise two coupled diffusion models to bidirectionally generate previous unobserved trajectories and future trajectories from random noises, which can address the issue of temporal information scarcity in limited observations. The underlying intuition behind this idea is: Both previous unobserved historical trajectories and future trajectories contain information of the same pedestrian at different timesteps, and thereby they provide complementary information to each other. It will be beneficial for the prediction of future trajectories if we can design an elegant method to simultaneously generate previous unobserved historical trajectories and future trajectories by fully leveraging the complementary information between them.

To accomplish this, we devise a step-by-step mutual guidance mechanism in two coupled diffusion models to simultaneously generate previous unobserved historical trajectories and future trajectories. Specifically, at each step, the predicted unobserved historical trajectories, together with the limited observed trajectories serve as a guidance for one diffusion model to predict a denoising intensity, which is then used to generate future trajectories of the subsequent step. Likewise, the predicted future trajectories, together with the observed trajectories, guide the other diffusion model to predict unobserved historical trajectories of the next step. Meanwhile, considering there exists relatively high noise in the generated trajectories during the initial steps, we devise a gating mechanism to learn the weights between the predicted trajectories and the limited observed trajectories for automatically controlling the proportion of the guidance information from two kinds of trajectories during each generation step. Through this iterative and mutually guided generation process, the future and unobserved historical trajectories are continuously refined, ultimately leading to precise predictions. Notably, our proposed BCDiff is encoder-free and is compatible with existing trajectory prediction encoders in principle, allowing them to gracefully handle cases with instantaneous observations.

Our contributions can be summarized as follows: 1) We propose BCDiff, a diffusion model based framework tailored for instantaneous trajectory prediction. BCDiff can simultaneously generate both future and unobserved historical trajectories in a consistent manner, which can effectively leverage complementary information between them. 2) We devise a step-by-step mutual guidance mechanism to couple two diffusion models for trajectory generation, and present a gating strategy to adaptively

adjust the contributions of the guidance information between two kinds of trajectories. 3) Experiments demonstrate our proposed BCDiff significantly improves the accuracy of instantaneous prediction and outperforms the state-of-the-art methods on ETH/UCY and Stanford Drone datasets.

## 2 Related Works

### 2.1 Traditional Trajectory Prediction

Traditional trajectory prediction methods aim to predict future trajectories given sufficient observation time. To capture complex interactions between pedestrians, many methods have been proposed [1, 15, 40]. These models utilize a social mechanism to aggregate neighboring actors and broadcast information to each actor. In addition, graph neural networks [49, 11, 19, 22, 27, 28] and transformer architectures [52, 36, 35, 51] are introduced to encapsulate implicit interactions among pedestrians. To resolve the problem of high uncertainty in pedestrians, researchers propose stochastic generative models, such as GAN [15, 22, 40, 45, 55], VAE [25, 24, 31, 53], and Diffusion Models [14, 32], to better capture the variability in future trajectories. Various sampling strategies are designed to avoid purely random sampling [6, 3, 18, 5, 30]. However, the aforementioned methods can accurately predict trajectories only when a sufficient amount of long observation trajectories are available. The accuracy cannot be guaranteed given the limited observation time. In contrast to these methods, our goal is to address the trajectory prediction problem under instantaneous observation scenarios.

### 2.2 Instantaneous Trajectory Prediction

Instantaneous trajectory prediction aims to predict future trajectories given a limited number of observed trajectory points. In the most extreme scenarios, merely two frames of locations can be observed. This task poses significant challenges due to the exceedingly short observation period. MOE [46] and DTO [34] are two trajectory prediction methods based on limited observed trajectory points. MOE proposes a feature extractor to incorporate image semantic information and develops a self-supervised task to enhance the representational ability of instantaneous observations. Meanwhile, DTO investigates the influence of noise introduced by detection or tracking to trajectory prediction, and intends to use fewer trajectory points in a knowledge distillation framework to address the issue. Despite these advances, they fail to address the inherent lack of temporal information in instantaneous trajectory prediction. In our study, we attempt to predict and leverage previous unobserved trajectories to capture more temporal information, thereby improving the prediction accuracy of future trajectories.

### 2.3 Diffusion Models

The diffusion model is a class of stochastic generation models, which exhibits amazing performance in a diverse range of fields such as image synthesis [9, 39, 33], audio synthesis [7, 21] and text generation [2, 12, 8]. Among these works, a typical diffusion model is the Denoising diffusion probabilistic model (DDPM) [44, 16]. DDPM is inspired by the non-equilibrium thermodynamics, in which a forward Markov process perturbs real data into noise, and a reverse Markov process converts noise back to real data. DDPM has been widely used in various tasks, including image super-resolution [41, 17], 3D point cloud generation [29, 56] etc. For instance, CDM [17] cascades multiple DDPM models to generate images of increasing resolution. The work in [29] employs a heat bath mechanism on DDPM to facilitate the generation of 3d point clouds. Different from these works, we leverage DDPM to solve the problem of instantaneous trajectory prediction.

## 3 Methods

### 3.1 Problem Formulation

In this work, we aim to tackle the task of instantaneous trajectory prediction, where we assume only two frames are observed, i.e., the most extreme case. We denote $X_{obs} = \{x_1, x_2\}$ as the observation locations. The ground-truth future trajectory is symbolized as $X_{fut} = \{x_3, x_4, ..., x_{T_{fut}+2}\}$, where $T_{fut}$ is the prediction length, and $x_i \in \mathbb{R}^2$ is the 2D coordinate of a trajectory location. Moreover, we characterize the previous unobserved trajectories as $X_{unobs} = \{x_{1-T_{unobs}}, ..., x_{-1}, x_0\}$, where $T_{unobs}$ represents the length of the unobserved trajectories. Our objective is to develop a diffusion

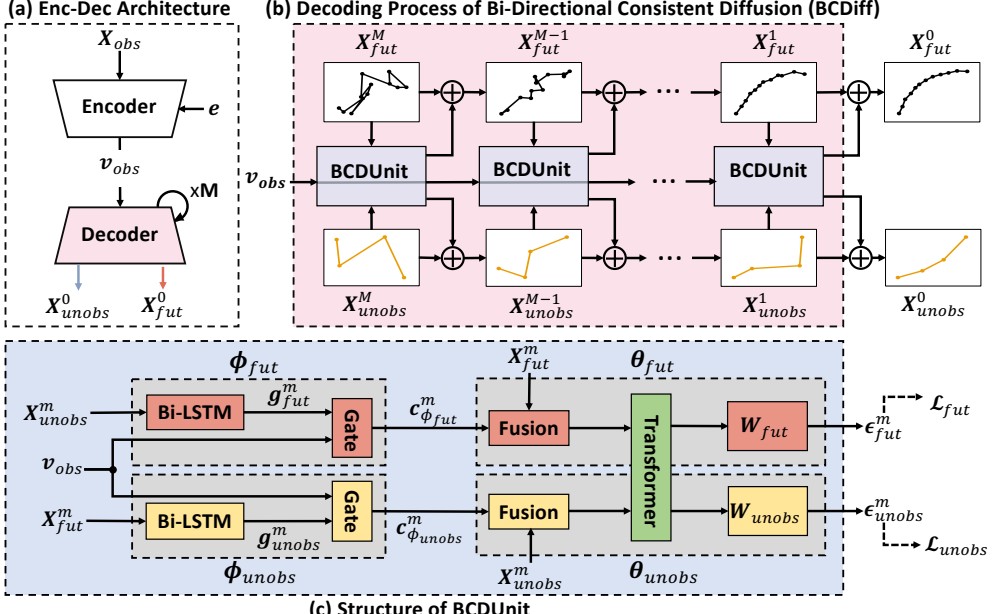

Figure 1: An illustration of our BCDiff framework. (a) The overall architecture comprises an encoder and a decoder. The encoder is utilized to generate features of trajectories by incorporating social and scene context. The decoder is depicted in Figure 1 (b). The decoder, i.e., our proposed BCDiff, generates previous unobserved historical trajectories and future trajectories step-by-step through two coupled diffusion models. (c) The BCDUnit describes the details of two diffusion models marked in red and yellow. We design a mutual guidance mechanism and predict denoise intensities that are used for generating unobserved historical trajectories and future trajectories at the next step.

models based method to generate both previous unobserved historical and future trajectories, so that more temporal information can be captured to better predict future trajectories, given only two observed frames. Since we utilize the diffusion model, we denote the maximum diffusion steps as $M$, and use $X_{fut}^m$ and $X_{unobs}^m$ to represent the future and unobserved trajectories following a diffusion of $m$ steps or a denoising of $M - m$ steps, respectively. Note that $X_{unobs} = X_{unobs}^0$ and $X_{fut} = X_{fut}^0$.

## 3.2 Overall Architecture

The overall architecture consists of an encoder and a decoder, as shown in Figure 1 (a). The encoder encodes $X_{obs}$ as $v_{obs}$, and captures social and scene context $\mathbf{e}$. The decoder is our proposed BCDiff framework, as shown in Figure 1 (b), which simultaneously generates previous observed historical trajectories and future trajectories step-by-step through the Bidirectional Consistent Denoising Unit (BCDUnit). The BCDUnit comprises two coupled diffusion models, As shown in Figure 1 (c), the diffusion model $\{\phi_{fut}, \theta_{fut}\}$ marked in red color, is responsible for generating future trajectories, while the diffusion model $\{\phi_{unobs}, \theta_{unobs}\}$, depicted in yellow, is used for generating unobserved historical trajectories. The two diffusion models are coupled through a mutual guidance mechanism. To be specific, as the $m^{th}$ step, the network $\phi = \{\phi_{fut}, \phi_{unobs}\}$ leverages two bidirectional LSTMs to encode unobserved trajectories and future trajectories as mutual guidance $g_{fut}^m$ and $g_{unobs}^m$ used for generating each other in the $m - 1^{th}$ step. Considering the guidance information contains relatively high noise during the initial steps, we introduce a gate mechanism to balance the contributions between the observed guidance $v_{obs}$ and future guidance $g_{fut}^m$, as well as the observed guidance $v_{obs}$ and unobserved guidance $g_{unobs}^m$, ultimately producing the appropriate guidance $\mathbf{c}_{\phi,fut}$ and $\mathbf{c}_{\phi,unobs}$. Then the network $\theta = \{\theta_{fut}, \theta_{unobs}\}$ fuses the guidance with both predicted unobserved historical and future trajectories at current steps to generate the denoise intensities, which is used to simultaneously generate $X_{fut}^{m-1}$ and $X_{unobs}^{m-1}$ of the next step. It is noteworthy that both previous unobserved historical trajectories and future trajectories contain the information of the same pedestrian at different timesteps, thus we adopt a parameter-shared transformer across $\theta_{fut}$ and $\theta_{unobs}$.

## 3.3 Bidirectional Consistent Diffusion

In this section, we introduce our proposed BCDiff framework, which contains two coupled diffusion models to consistently generate trajectories in two directions, i.e., simultaneously predicting previous unobserved trajectories and future trajectories. In this paper, we utilize DDPM [16] as our basic diffusion model, because of its excellent performance in various tasks. For the generation of each direction, the diffusion model executes diffusion and conditional denoising processes. The diffusion process aims to intentionally add a series of noises to a ground-truth trajectory, while the conditional denoising process recovers the trajectory from noise inputs conditioned on the guidance.

**Diffusion Process.** The diffusion process is defined as a Markov chain, conditioned on the ground-truth trajectories $X^0_{fut}, X^0_{unobs}$. To write conveniently, we omit the subscripts, allowing $X^0$ to represent unobserved historical trajectories or future trajectories. The diffusion process generates the sequence $\{X^i\}^M_{i=1}$ by accumulating noise $M$ times, i.e.,

$$q(X^{1:M}|X^0) = \prod_{m=1}^{M} q(X^m|X^{m-1}), \quad q(X^m|X^{m-1}) = \mathcal{N}(X^m; \sqrt{\alpha^m}X^{m-1}, (1-\alpha^m)\mathbf{I}), \quad (1)$$

where $\mathcal{N}$ denotes the Gaussian distribution, and $\alpha^m$ represents the noise intensity from $X^m$ to $X^{m-1}$. Typically, $\alpha^m$ is equal to $1 - \beta^m$, where $\beta^m$ is a pre-defined value belonging to the interval [0,1]. Due to the additivity of Gaussian distributions, we are able to directly obtain $X^m$ from $X^0$:

$$q(X^m|X^0) = \mathcal{N}(X^m; \sqrt{\overline{\alpha}^m}X^0, (1-\overline{\alpha}^m)\mathbf{I}), \quad \overline{\alpha}^m = \prod_{i=1}^{m} \alpha^i, \quad (2)$$

where $\overline{\alpha}^m = \prod_{i=1}^{m} \alpha^i$. Note that as $M$ becomes sufficiently large, $\overline{\alpha}$ approaches to zero, making $q(X^M|X^0)$ converge to the standard Gaussian distribution. By employing the above process, the ground-truth trajectory is transformed into the Gaussian noise $X^M \sim \mathcal{N}(0, I)$.

**Conditional Denoising Process.** In this process, we aim to generate $X^0$ from the Gaussian noise $X^M \sim \mathcal{N}(0, I)$. We can reverse the aforementioned diffusion process, to gradually denoise from the Gaussian noise $X^M$ and reconstruct $X^0$. Based on the proof in [10]: If $q(X^m|X^{m-1})$ follows a Gaussian distribution and $\beta^m$ is sufficiently small, $q(X^{m-1}|X^m)$ also satisfies a Gaussian distribution. Therefore, we formulate $q(X^{m-1}|X^m)$ as a Gaussian Markov process. However, it is intractable to directly obtain $q(X^{m-1}|X^m)$. Consequently, we employ a denoising neural network to estimate its mean and variance:

$$p_\Theta(X^{0:M}|\mathbf{c}_\phi^m) = p(X^M|\mathbf{c}_\phi^m) \prod_{m=1}^{M} p_\Theta(X^{m-1}|X^m, \mathbf{c}_\phi^m), \quad (3)$$

$$p_\Theta(X^{m-1}|X^m, \mathbf{c}_\phi^m) = \mathcal{N}(X^{m-1}|\mathbf{c}_\phi^m; \mu_\Theta(X^m, m, \mathbf{c}_\phi^m), \mathbf{\Sigma}_\Theta(X^m, m, \mathbf{c}_\phi^m)), \quad (4)$$

where $\Theta = \{\phi, \theta\}$ is the parameter of the neural network. $\mu_\Theta$ and $\Sigma_\Theta$ are the predicted mean and variance by $\Theta$. $\mathbf{c}_\phi^m$ is produced by network $\phi$, serving as conditions to guide the denoising (we will introduce it in detail later). Our ultimate goal in the denoising process is to ensure that the denoising step becomes the inverse process of the diffusion step, thus enabling $p_\Theta(X^{m-1}|X^m, \mathbf{c}_\phi^m)$ and $q(X^{m-1}|X^m)$ to have the same distribution. For more details, please refer to Section 3.4.

**Bidirectional Consistent Denoising Unit.** To utilize more temporal complementary information in trajectories, we propose the BCDUnit to couple two diffusion models. One is called the backward model, used to generate previous unobserved historical trajectories. The other, named as the forward model, is employed to produce future trajectories. We design a mutual guidance mechanism in the two diffusion models: At each step, the predicted unobserved historical trajectories and observed trajectories are jointly utilized to guide the forward model to generate future trajectories of the next step. Likewise, the predicted future trajectories together with observed trajectories are responsible for guiding the backward model to generate the unobserved historical trajectories. In this way, the BCDUnit continuously refines the future and unobserved historical trajectories, ultimately leading to accurate predictions.

We denote the network in BCDUnit as $\Theta = \{\{\phi_{fut}, \theta_{fut}\}, \{\phi_{unobs}, \theta_{unobs}\}\}$. As illustrated in Figure 1 (c), the forward model $\{\phi_{fut}, \theta_{fut}\}$, marked as red, is used to generate future trajectories, while the backward model $\{\phi_{unobs}, \theta_{unobs}\}$, depicted in yellow, is responsible for generating

unobserved historical trajectories. Here, the guidance information can be obtained by the network $\phi = \{\phi_{fut}, \phi_{unobs}\}$. At the $m^{th}$ step, $\phi$ first generates future guidance $g_{fut}^m$ and unobserved guidance $g_{unobs}^m$ by sending the unobserved historical trajectories $X_{unobs}^m$ and future trajectories $X_{fut}^m$ to Bidirectional Long-Short Term Memory (Bi-LSTM), respectively:

$$g_{fut}^m = \textbf{Bi-LSTM}(X_{unobs}^m), \; g_{unobs}^m = \textbf{Bi-LSTM}(X_{fut}^m). \tag{5}$$

We further incorporate observed trajectory guidance $v_{obs}$ into future and unobserved guidance to obtain more informative guidance $\mathbf{c}_{\phi,fut}^m$ and $\mathbf{c}_{\phi,unobs}^m$, $\mathbf{c}_{\phi,fut}^m = [g_{fut}^m, v_{obs}, m]$, $\mathbf{c}_{\phi,unobs}^m = [g_{unobs}^m, v_{obs}, m]$, where $[\cdot, \cdot]$ represents the concatenation operation. However, considering that it contains relatively high noise in the initial steps due to the inherent property of the diffusion model, it is not appropriate to directly concatenate them in the beginning. To this end, we adopt a gating mechanism to automatically learn the weights for balancing the contributions between two kinds of guidance information. We first calculate the weights for the future guidance $g_{fut}^m$ and observed trajectory guidance $v_{obs}$ to produce appropriate guidance. Formally,

$$\gamma_{fut}^m = \textbf{Gate}_{fut}([g_{fut}^m, v_{obs}, m]), \tag{6}$$

where $\gamma_{fut}^m$ is the learnt weight. $\textbf{Gate}$ is a two layers MLP with the Sigmoid activation in this paper. The guidance $\mathbf{c}_{\phi,fut}^m$ can be then obtained by:

$$\mathbf{c}_{\phi,fut}^m = \gamma_{fut}^m g_{fut}^m + (1 - \gamma_{fut}^m)v_{obs}. \tag{7}$$

Similarly, $\mathbf{c}_{\phi,unobs}^m$ can be obtained in the same way. After obtaining $\mathbf{c}_{\phi,fut}^m$ and $\mathbf{c}_{\phi,unobs}^m$, the network $\theta = \{\theta_{fut}, \theta_{unobs}\}$ utilizes them to perform mutual guidance. They fuse the guidance with the predicted trajectories at the $m^{th}$ step. Then, a transformer is leveraged to capture temporal dependencies in the fused features. Finally, the outputs of the transformer are passed through two MLPs, i.e., $W_{fut}$ and $W_{unobs}$, to obtain denoising intensities $\epsilon_{fut}^m$ and $\epsilon_{unobs}^m$, respectively. These denoising intensities are employed to generate trajectories of the next steps, *i.e.*, $X_{fut}^{m-1}$ and $X_{unobs}^{m-1}$.

In this way, we can bidirectionally and consistently generate unobserved historical trajectories and future trajectories step-by-step, effectively utilizing the complementary information between them.

### 3.4 The Objective Function

We define the objective as the negative log-likelihood of the model $p_\Theta$ under $X_{fut}^0$ and $X_{unobs}^0$ as

$$\mathcal{L} = \mathbb{E}[-log p_\Theta(X^0)]. \tag{8}$$

Here, we also omit the subscript for convenient writing. By minimizing the objective $\mathcal{L}$, the original trajectories $X_{fut}^0$, and $X_{unobs}^0$ can be recovered through the denoising process. However, it is difficult to directly compute $\mathcal{L}$. Therefore, we employ the variational methods to derive the Variational Lower Bound (VLB) [20] of the expectation, denoted as:

$$\mathcal{L} \leq -\mathcal{L}_{VLB} = \mathbb{E}_q\left[log\frac{q(X^{1:M}|X^0)}{p_\Theta(X^{0:M})}\right] = \mathbb{E}_q[KL(q(X^M|X^0)||p_\Theta(X^M|\mathbf{c}_\phi^M)$$

$$- log p_\Theta(X^0|X^1, \mathbf{c}_\phi^1) + \sum_{m=2}^M KL(q(X^{m-1}|X^m, X^0)||p_\Theta(X^{m-1}|X^m, \mathbf{c}_\phi^m))].$$

The first term of $\mathcal{L}_{VLB}$ approximates to 0, as both $q(X^M|X^0)$ and $p_\Theta(X^M|\mathbf{c}_\phi^M)$ are approximate to $\mathcal{N}(0, I)$. The second term can be formulated as a special case of the third term when $m = 1$. The third term computes the KL divergence between the estimated distribution $p_\Theta(X^{m-1}|X^m, \mathbf{c}_\phi^m)$ and the true posterior distribution $q(X^{m-1}|X^m, X^0)$, aiming to lower the error between the estimated distribution and the ground-truth posterior distribution. To determine the $q(X^{m-1}|X^m, X^0)$, we apply the Bayes formula as follows:

$$q(X^{m-1}|X^m, X^0) = q(X^m|X^{m-1}, X^0)\frac{q(X^{m-1}|X^0)}{q(X^m|X^0)} = q(X^m|X^{m-1})\frac{q(X^{m-1}|X^0)}{q(X^m|X^0)}. \tag{9}$$

By applying the Bayes formula, we observe each term can be calculated with Equation 2. We then substitute the results of Equation 2 to Equation 9 to obtain the mean and variance of the posterior distribution $q(X^{m-1}|X^m, X^0)$ as:

$$\widetilde{\sigma}^m = \frac{1 - \overline{\alpha}^{m-1}}{1 - \overline{\alpha}^m} \cdot \beta^m, \quad \widetilde{\mu}^m(X^m, X^0) = \frac{\sqrt{\alpha^m}(1 - \overline{\alpha}^{m-1})}{1 - \overline{\alpha}^m}X^m + \frac{\sqrt{\overline{\alpha}^{m-1}}\beta^m}{1 - \overline{\alpha}^m}X^0. \tag{10}$$

Note that $\widetilde{\sigma}^m$ is a constant value related to $\beta^m$, thus the KL in the third term can be further derived as:

$$KL(q(X^{m-1}|X^m, X^0)||p_\Theta(X^{m-1}|X^m, \mathbf{c}_\phi^m)) \propto ||\widetilde{\mu}^m(X^m, X^0) - \mu_\Theta(X^m, m, \mathbf{c}_\phi^m)||_2. \tag{11}$$

By reparameterizing $\mu_\Theta$ and substituting it into Equation 11, we can further simplify the expression and finally obtain the diffusion loss.

$$\mathcal{L}_{unobs} = \mathbb{E}_{X_{unobs}^0, \epsilon, m}||\epsilon - \epsilon_\Theta(X_{unobs}^m, m, \mathbf{c}_{\phi,unobs}^m)||_2$$
$$\mathcal{L}_{fut} = \mathbb{E}_{X_{fut}^0, \epsilon, m}||\epsilon - \epsilon_\Theta(X_{fut}^m, m, \mathbf{c}_{\phi,fut}^m)||_2$$
$$\mathcal{L}_d = \mathcal{L}_{fut} + \mathcal{L}_{unobs} \tag{12}$$

where $\epsilon \sim \mathcal{N}(0, I)$, $X_{fut}^m$ and $X_{unobs}^m$ are calculated by Equation 2, and $\epsilon_\Theta$ represents the aforementioned BCDUnit.

### 3.5 Mutual Guidance based Optimizing and Inference

When optimizing the model, we take the following four steps: Firstly, we sample future ground-truth trajectories $X_{fut}^0$, instantaneous observations $X_{obs}$, and unobserved historical ground-truth trajectories $X_{unobs}^0$ from the training dataset. We also sample a timestep $m \sim uniform(1, M)$ and a Gaussian noise $\epsilon \sim \mathcal{N}(0, I)$. Secondly, we use the encoder to encode $X_{obs}$ as the $v_{obs}$, and employ the Equation 2 to obtain the $X_{fut}^m$ and $X_{unobs}^m$. Thirdly, we utilize the Equation 5,6, and 7 to obtain mutual guidance $\mathbf{c}_{\phi,unobs}^m$ and $\mathbf{c}_{\phi,fut}^m$. Finally, we calculate the loss defined in Equation 12 and take the gradient descent to optimize the model until it converges.

For inference, we first sample instantaneous observations $X_{obs}$ from the testing data and two Gaussian noises $\hat{X}_{fut}^M$ and $\hat{X}_{unobs}^M$ from $\mathcal{N}(0, I)$. Following the third step in the optimizing stage, we obtain mutual guidance $\mathbf{c}_{\phi,fut}^M$ and $\mathbf{c}_{\phi,unobs}^M$. Finally, we execute the following two updates from $m = M$ to 1 so that the predicted trajectories can be continuously refined until $\hat{X}_{fut}^0$ and $\hat{X}_{unobs}^0$ are generated,

$$\hat{X}_{fut}^{m-1} := \frac{1}{\sqrt{\alpha^m}}(\hat{X}_{fut}^m - \frac{\beta^m}{\sqrt{1 - \overline{\alpha}^m}}\epsilon_\Theta(\hat{X}_{fut}^m, m, \mathbf{c}_{\phi,fut}^m)) + \widetilde{\sigma}^m\epsilon, \tag{13}$$

$$\hat{X}_{unobs}^{m-1} := \frac{1}{\sqrt{\alpha^m}}(\hat{X}_{unobs}^m - \frac{\beta^m}{\sqrt{1 - \overline{\alpha}^m}}\epsilon_\Theta(\hat{X}_{unobs}^m, m, \mathbf{c}_{\phi,unobs}^m)) + \widetilde{\sigma}^m\epsilon, \tag{14}$$

where $\epsilon$ is a random noise sampled from the standard Gaussian distribution. The details of training and inference procedures are provided in Appendix.

## 4 Experiments

### 4.1 Experiment Settings

**Dataset.** We verify the effectiveness of our proposed method on the widely used ETH/UCY [37, 26] and Stanford Drone [38] Dataset (SDD). ETH/UCY is a dataset group. It consists of 5 different scenes, among which 2 scenes (ETH, HOTEL) are from the ETH dataset, and the other three scenes (UNIV, ZARA1, and ZARA2) come from the UCY dataset. The whole dataset includes more than 1500 pedestrians. We follow the widely used leave-one-scene-out protocol, i.e., the models are trained on 4 scenes and tested on the remaining one [15, 42]. SDD is a large scale dataset consisting of 20 scenes. It contains various agents such as pedestrians, bicycles, and vehicles.

**Evaluation Metrics.** Following previous works [43, 19, 14, 53, 48], we employ the Average Displacement Error (ADE) and Final Displacement Error (FDE) as metrics to evaluate the performance of future trajectory predictions. In the instantaneous trajectory prediction setting, the observations are

Table 1: Comparisons of different methods on the ETH/UCY dataset. The metrics are presented as ADE/FDE (m).

| Model | Methods | Dataset | | | | | |
| --- | --- | --- | --- | --- | --- | --- | --- |
| | | ETH | HOTEL | UNIV | ZARA1 | ZARA2 | AVG |
| Trajectron++ | Instantaneous | 0.76/1.43 | 0.30/0.56 | 0.36/0.74 | 0.22/0.42 | 0.18/0.34 | 0.36/0.70 |
| | MOE [46] | 0.64/1.12 | 0.20/0.33 | 0.33/0.62 | 0.22/0.42 | **0.17/0.32** | 0.31/0.56 |
| | MOE w/o Image | 0.68/1.22 | 0.25/0.49 | 0.35/0.68 | 0.22/0.42 | 0.18/0.33 | 0.34/0.63 |
| | DTO [34] | 0.70/1.23 | 0.22/0.45 | 0.32/0.62 | 0.22/0.42 | **0.17**/0.33 | 0.33/0.61 |
| | BCDiff | **0.61/1.09** | **0.16/0.28** | **0.28/0.53** | **0.22/0.41** | 0.18/0.33 | **0.29/0.53** |
| PCCSNet | Instantaneous | 0.34/0.65 | 0.14/0.25 | 0.31/0.63 | 0.23/0.46 | 0.16/0.37 | 0.24/0.47 |
| | MOE [46] | 0.31/0.57 | **0.13**/0.21 | **0.25**/0.53 | 0.20/0.41 | **0.14/0.31** | 0.20/0.41 |
| | MOE w/o Image | 0.32/0.61 | 0.14/0.24 | 0.28/0.57 | 0.21/0.45 | 0.15/0.34 | 0.22/0.44 |
| | DTO [34] | 0.33/0.64 | 0.14/0.24 | 0.31/0.62 | 0.22/0.46 | 0.15/0.35 | 0.23/0.46 |
| | BCDiff | **0.30/0.56** | **0.13/0.20** | **0.25/0.52** | **0.18/0.37** | **0.14/0.31** | **0.19/0.39** |
| SGCN | Instantaneous | 0.88/1.66 | 0.55/1.16 | 0.38/0.71 | 0.30/0.54 | 0.25/0.46 | 0.47/0.91 |
| | MOE [46] | 0.74/1.41 | 0.45/0.85 | 0.38/0.71 | **0.29**/0.54 | 0.25/0.45 | 0.42/0.79 |
| | MOE w/o Image | 0.79/1.52 | 0.50/1.05 | 0.38/0.71 | 0.30/0.54 | 0.25/0.46 | 0.44/0.85 |
| | DTO [34] | 0.80/1.56 | 0.49/1.02 | 0.38/0.71 | 0.30/0.54 | 0.25/0.46 | 0.44/0.86 |
| | BCDiff | **0.66/1.18** | **0.34/0.62** | **0.38/0.70** | 0.30/**0.54** | **0.25/0.44** | **0.39/0.72** |
| SocialVAE | Instantaneous | 0.64/1.10 | 0.21/0.34 | 0.27/0.51 | 0.22/0.39 | 0.18/0.34 | 0.30/0.54 |
| | MOE [46] | 0.57/1.01 | 0.17/0.29 | 0.26/0.44 | 0.22/**0.36** | 0.17/0.32 | 0.28/0.48 |
| | MOE w/o Image | 0.59/1.06 | 0.20/0.33 | 0.27/0.49 | 0.22/0.38 | 0.18/0.33 | 0.29/0.52 |
| | DTO [34] | 0.61/1.06 | 0.18/0.31 | 0.25/0.43 | 0.22/0.38 | 0.17/0.33 | 0.29/0.50 |
| | BCDiff | **0.53/0.91** | **0.17/0.27** | **0.24/0.40** | **0.21**/0.37 | **0.16/0.26** | **0.26/0.44** |

reduced to 2 frames, and the length of future predictions is 12 frames. Following previous works [46, 15, 31], we sample 20 future predicted trajectories, and report the final error by the minimum error over all predicted trajectories.

**Backbone and Baselines.** To demonstrate the compatible ability of our BCDiff, we apply it to three popular trajectory prediction models, Trajectron++ [42], PCCSNet [47], SGCN [43], and SocialVAE [50] by replacing their decoders with our BCDiff. Moreover, we compare BCDiff with the following baselines: **Instantaneous** means directly predicting the trajectories of the next 12 frames conditions on 2 frames of observations using the above three backbones. Additionally, we take **MOE** [46] and **DTO** [34] as two baselines for instantaneous trajectory prediction. Since the original MOE employs image semantic information, we also implement MOE without using image semantic information, denoted as **MOE w/o Image**, in order to fairly compare with other methods.

## 4.2 Experiment Results and Analysis

**Performance on Instantaneous Trajectory Prediction.** The overall performance are listed in Table 1 and 2. The results by applying our BCDiff to three different backbones, consistently outperform the baseline methods on all the datasets. This demonstrates the effectiveness of our proposed method for instantaneous trajectory prediction. Meanwhile, it also illustrates our method can be well compatible with different trajectory prediction models. Note that the performance of MOE declines when image information is not included, which shows that MOE heavily depends on image semantic information. In contrast, our proposed method does not rely on any additional image information.

**Ablation Study.** We perform the ablation studies, as listed in Table 3. We first utilize two separate diffusion models to predict previous unobserved trajectories and future trajectories, respectively, denoting it as BCDiff-w/o.Guidance. Then, we only utilize unobserved and observed trajectories to guide the generation of future trajectories, denoted as BCDiff-Uni.Guidance. BCDiff-Uni.Guidance is better than BCDiff-w/o.Guidance, meaning that the guidance from previous unobserved historical trajectories is helpful for improving the prediction performance of future trajectories. Moreover, by

Table 2: Comparisons of different methods on the Stanford Drone dataset. The metrics are presented as ADE/FDE (m).

| Methods | Trajectron++ | PCCSNet | SGCN | SocialVAE |
|---|---|---|---|---|
| Instantaneous | 13.07/22.88 | 9.19/17.71 | 15.40/25.69 | 9.56/16.10 |
| MOE[46] | 11.71/19.54 | 8.40/16.08 | 14.45/24.88 | 9.12/14.98 |
| MOE w/o Image | 12.41/21.46 | 8.87/16.80 | 15.02/25.13 | 9.39/15.87 |
| DTO[34] | 12.32/20.79 | 8.93/16.92 | 14.99/25.07 | 9.28/15.58 |
| BCDiff | **11.56/19.32** | **8.32/15.87** | **13.67/23.92** | **9.05/14.86** |

Table 3: Ablation Studies using Trajectron++ on the ETH/UCY and Stanford Drone datasets.

| Method | ETH/UCY | SDD |
|---|---|---|
| BCDiff-w/o.Guidance | 0.36/0.70 | 13.07/22.88 |
| BCDiff-Uni.Guidance | 0.33/0.60 | 12.57/21.26 |
| BCDiff-Bi.Guidance | 0.32/0.57 | 11.96/19.88 |
| BCDiff-Bi.Guidance & Gate | 0.29/0.53 | 11.56/19.32 |

incorporating bidirectional guidance, called BCDiff-Bi.Guidance, we observe the performance is further improved. This illustrates the guidance from future trajectories is beneficial for predicting previous unobserved trajectories, thereby improving the final prediction performance. Finally, we integrate the gating mechanism into our model, named as BCDiff-Bi.Guidance & Gate. It achieves the best performance, showing the gating mechanism is effective for trajectory prediction.

**Analysis on Length of Predicted Unobserved Points.** We investigate the impact of the number of predicted unobserved points on trajectory prediction. We use the Trajectron++ encoder [42], as listed in Table 4. As the number of predicted points increases, the accuracy of future predictions gradually improves, which can be attributed to the additional unobserved points providing more useful information. When $T = 5$, the prediction performance starts to decline. This is because it becomes unprecise when predicting previous unobserved historical points with a larger length, thus introducing noise into our method.

Table 4: Analysis of different unobserved point lengths $T_{unobs}$

| Dataset | Direction | $T_{unobs} = 1$ | $T_{unobs} = 2$ | $T_{unobs} = 3$ | $T_{unobs} = 4$ | $T_{unobs} = 5$ | $T_{unobs} = 6$ |
|---|---|---|---|---|---|---|---|
| ETH/UCY | Future | 0.31/0.55 | 0.30/0.54 | 0.30/0.53 | **0.29/0.53** | 0.32/0.57 | 0.34/0.60 |
| | Unobserved | 0.006/0.006 | 0.034/0.033 | 0.055/0.063 | 0.086/0.089 | 0.108/0.133 | 0.149/0.212 |
| SDD | Future | 11.86/19.78 | 11.81/19.71 | 11.68/19.50 | **11.56/19.32** | 11.92/20.01 | 12.33/20.98 |
| | Unobserved | 0.311/0.312 | 1.564/1.492 | 2.626/2.678 | 3.552/3.884 | 4.828/5.423 | 6.177/7.894 |

**Qualitative Analysis.** We visualize the predicted trajectories in four different scenarios: Walking side-by-side, walking along, turning and yielding, as shown in Figure 2. BCDiff can accurately predict future trajectories, compared to MOE and DTO in all four scenarios. This is because BCDiff utilizes temporal information of unobserved trajectories to aid the prediction of future trajectories. Note that when pedestrians are walking side-by-side, as depicted in the first column of Figure 2, BCDiff utilizes the predicted unobserved historical trajectories to capture the walking patterns between two pedestrians, thereby precisely forecasting future trajectories. However, DTO and MOE predict deviated trajectories, due to only using two observed frames. Moreover, we visualize

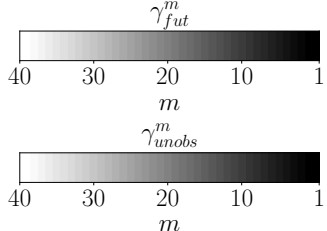

Figure 3: Visualization of weights $\gamma_{fut}^m$ and $\gamma_{unobs}^m$. Darker colors represent larger values.

the gating weights $\gamma_{fut}^m$ and $\gamma_{unobs}^m$ in the denoising process from $m = 40$ to $m = 1$. As shown in Figure 3, the weights are gradually increased. It indicates our gating mechanism can adaptively learn weights, assigning lower weights to the predicted guidance in the initial steps.

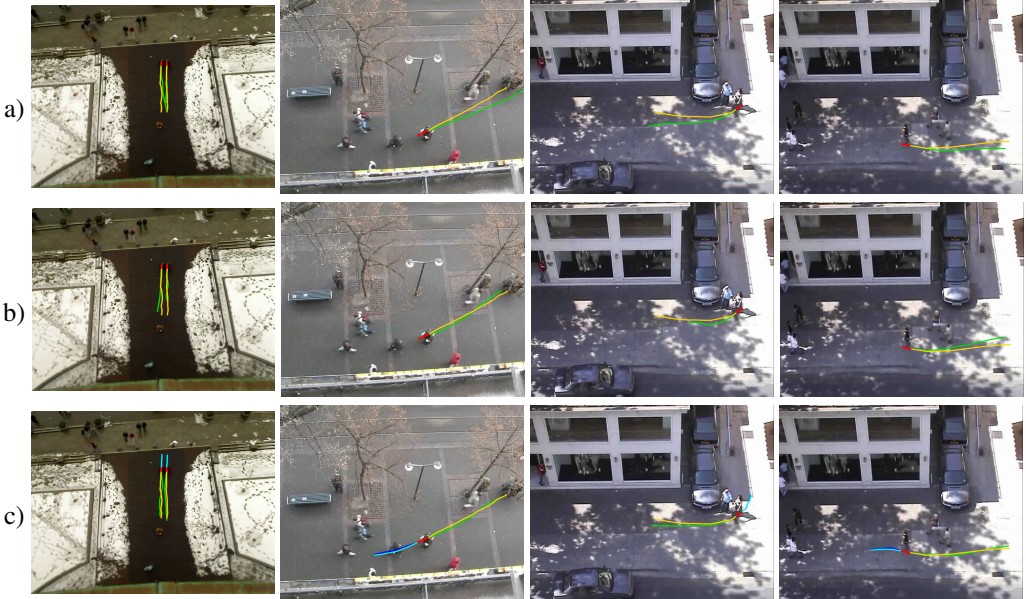

Figure 2: Visualization of predicted trajectories on the ETH/UCY Dataset. Given the instantaneous observed trajectories (red), we predict the future trajectories (green) by (a) MOE, (b) DTO and (c) Our BCDiff. The ground-truth future trajectories are shown in orange color. In addition, we also draw the predicted unobserved trajectories (blue) by our BCDiff and ground-truth (cyan). Our predicted future trajectories are closer to the ground-truth, compared to other methods.

## 5 Conclusion

We proposed a diffusion model based framework for the task of instantaneous trajectory prediction. The proposed framework simultaneously generated both unobserved historical and future trajectories by designing a mutual guidance mechanism to couple two diffusion models, such that more temporal information can be leveraged for future trajectory prediction. We introduced a gating strategy, automatically balancing the contributions of different guidance information. Experiments demonstrated our proposed framework achieved superior performance to the state-of-the-art methods.

## Acknowledgments and Disclosure of Funding

This work was supported by the NSFC under Grants 62122013, U2001211. This work was also supported by the Innovative Development Joint Fund Key Projects of Shandong NSF under Grants ZR2022LZH007.

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
