# A Implementation details

**Hyperparameters and Network structures.** We set the diffusion steps of both diffusion models to $M = 40$. The value of $\beta$ ranges from 1e-4 to 2e-1, linearly increasing with the number of diffusion steps. The dimension of the encoded $v_{obs}$ is 256. We use Adam [6] as the optimizer, where the learning rate and the batch size are set to 1e-3 and 256, respectively. All the experiments are performed on a NVIDIA RTX 3090 GPU, PyTorch 1.11.0 platform [5]. For the network $\phi$ of BCDUnit, we set the hidden size of Bi-LSTM to 256, and set the output dimensions of two layers MLP in **Gate** to 128 and 1. Additionally, for the network $\theta$, the fusion module combines the predicted trajectories with the guidance features through

$$H^m_{fut} = (W^1_{fut}X^m_{fut} + b^1_{fut}) \otimes (W^2_{fut}\mathbf{c}^m_{fut} + b^2_{fut}) + (W^3_{fut}\mathbf{c}^m_{\phi,fut} + b^3_{fut})$$

$$H^m_{unobs} = (W^1_{unobs}X^m_{unobs} + b^1_{unobs}) \otimes (W^2_{unobs}\mathbf{c}^m_{unobs} + b^2_{unobs}) + (W^3_{unobs}\mathbf{c}^m_{\phi,unobs} + b^3_{unobs}),$$

where $\otimes$ represents the element-wise product. $\{W^i_{fut}\}^3_{i=1}$ and $\{b^i_{fut}\}^3_{i=1}$ are trainable weights for the $\theta_{fut}$. $\{W^i_{unobs}\}^3_{i=1}$ and $\{b^i_{unobs}\}^3_{i=1}$ are trainable weights for the $\theta_{unobs}$. Both of the dimensions of $H^m_{fut}$ and $H^m_{unobs}$ are set to 256. We use a share-parameter Transformer for both $\theta_{fut}$ and $\theta_{unobs}$, which consists of 2 layers with dimensions of 512 and 4 attention heads. For both of $W_{fut}$ and $W_{unobs}$, we utilize 3 layers MLP whose output sizes are 256, 128 and 2.

The social context $\mathbf{e}$ is the information of other moving pedestrians around the target pedestrian whose future trajectories are to be predicted. The scene context represents the map information around the target agent. Generally speaking, these two kinds of information will be integrated into the encoder of the trajectory prediction model to boost the representation ability of historical trajectories. Since our proposed BCDiff is actually a decoder, in other words, it is an encoder-agnostic framework, our attention is not paid to the contextual information.

**Baseline Implementation.** We introduce how to compare baselines with our proposed method in Table 1 and 2. Since all MOE [7], DTO [8] and BCDiff are plug-to-play methods, we apply them to three kind of prediction backbones (Trajectron++, PCCSNet and SGCN). In the case of MOE, following the approach described in their paper [7], we replace the encoder of the backbone with their proposed MOE encoder. Regarding DTO, we initially pre-train a backbone model using 8 historical observations. Subsequently, we distill a student model using only 2 historical observations for instantaneous trajectory prediction. For BCDiff, we replace the decoder of the backbone with our BCDiff framework.

# B Limitations

Although BCDiff demonstrates promising performance in instantaneous trajectory prediction, it requires approximately 4 seconds for predicting trajectories once, which is somehow time-consuming. This is primarily due to the refinement process of both unobserved historical trajectories and future trajectories through BCDUnit, which is iterated 40 times (with $M = 40$). However, it is worth noting that there might be some potentially feasible acceleration ways:

**From the perspective of diffusion algorithms:** Our current framework is built based on a typical diffusion model, DDPM. DDPM models the trajectory generation process as a Markov process that requires continuous M-steps to generate trajectories. There are many cutting-edged methods for accelerating inference [1, 2, 3, 4], by relaxing the constraints of the Markov process and generating trajectories using skipping steps. In this way, the model can use less steps to generate a trajectory, and thereby shorten the reference time.

**From the perspective of model structure:** We can combine our method with Half-Precision/Model Quantization techniques [11, 12] to lower the computational cost, or Model Pruning [9, 10] to simplify the model, or Knowledge Distillation [13, 14] to utilize a small but effective model for inference.

# C Training and Inference Procedure of BCDiff

We present the training and inference procedures for BCDiff in Algorithm 1 and Algorithm 2, respectively.

**Algorithm 1:** Training Procedure of BCDiff

**while** *Model not converges* **do**
> Sample trajectory $(X_{fut}^0, X_{obs}, X_{unobs}^0)$ from dataset
> Sample $m \sim \text{Uniform}(1, M)$
> Sample $\epsilon \sim \mathcal{N}(0, I)$
> Encode $X_{obs}$ as $v_{obs}$
> Employ the Equation 2 to obtain the $X_{fut}^m$ and $X_{unobs}^m$
> Calculate guidance $\mathbf{c}_{\phi,fut}^m$ and $\mathbf{c}_{\phi,unobs}^m$ through Equation 5, 6, and 7:
> $$\mathcal{L}_d = \mathbb{E}_{X_{fut}^0, \epsilon, m} ||\epsilon_{fut} - \epsilon_\Theta(X_{fut}^m, m, \mathbf{c}_{\phi,fut}^m)||_2$$
> $$\quad + \mathbb{E}_{X_{unobs}^0, \epsilon, m} ||\epsilon_{unobs} - \epsilon_\Theta(X_{unobs}^m, m, \mathbf{c}_{\phi,unobs}^m)||_2$$
> Calculate the gradient $\nabla\mathcal{L}_d$ and take gradient descent to update the whole model

**end**

---

**Algorithm 2:** Inference Procedure of BCDiff

**Input:** observed trajectories $X_{obs}$
**Output:** Predicted unobserved trajectories $\hat{X}_{unobs}$, predicted future trajectories $\hat{X}_{fut}$
Sample $\hat{X}_{unobs}^M, \hat{X}_{fut}^M \sim \mathcal{N}(0, I)$
Encode $X_{obs}$ as $v_{obs}$
**for** $m = M, ..., 1$ **do**
> sample $\epsilon \sim \mathcal{N}(0, I)$ if m > 1, else $\epsilon = 0$
> Calculate guidance $\mathbf{c}_{\phi,fut}^m$ and $\mathbf{c}_{\phi,unobs}^m$ through Equation 5, 6, and 7
> Generate trajectories of next step through:
> $$\hat{X}_{unobs}^{m-1} = \frac{1}{\sqrt{\alpha^m}}(\hat{X}_{unobs}^m - \frac{\beta^m}{\sqrt{1-\overline{\alpha}^m}}\epsilon_\Theta(\hat{X}_{unobs}^m, m, \mathbf{c}_{\phi,unobs}^m)) + \widetilde{\sigma}^m \epsilon_{unobs}$$
> $$\hat{X}_{fut}^{m-1} = \frac{1}{\sqrt{\alpha^m}}(\hat{X}_{fut}^m - \frac{\beta^m}{\sqrt{1-\overline{\alpha}^m}}\epsilon_\Theta(\hat{X}_{fut}^m, m, \mathbf{c}_{\phi,fut}^m)) + \widetilde{\sigma}^m \epsilon_{fut}$$

**end**
$\hat{X}_{unobs} = \hat{X}_{unobs}^0, \ \hat{X}_{fut} = \hat{X}_{fut}^0$

# D  Derivation of $X^m$ from $X^0$

We derive the process from $X^0$ to $X^m$ in Equation 2. Recall that we have

$$q(X^m|X^{m-1}) = \mathcal{N}(X^m; \sqrt{\alpha^m}X^{m-1}, (1-\alpha^m)\mathbf{I}).$$

By using reparameterization, we obtain,

$$\begin{aligned}
X^m &= \sqrt{\alpha^m}X^{m-1} + \sqrt{1-\alpha^m}\epsilon^{m-1} \text{ ;where } \epsilon^m \text{and } \epsilon^{m-1} \sim \mathcal{N}(0, I)\\
&= \sqrt{\alpha^m}(\sqrt{\alpha^{m-1}}X^{m-2} + \sqrt{1-\alpha^{m-1}}\epsilon^{m-2}) + \sqrt{1-\alpha^m}\epsilon^{m-1}\\
&= \sqrt{\alpha^m\alpha^{m-1}}X^{m-2} + \sqrt{\alpha^m(1-\alpha^{m-1})}\epsilon^{m-2} + \sqrt{1-\alpha^m}\epsilon^{m-1} \quad (15)\\
&= \sqrt{\alpha^m\alpha^{m-1}}X^{m-2} + \sqrt{1-\alpha^m\alpha^{m-1}}\overline{\epsilon}^{m-2} \text{ ;where } \overline{\epsilon}^{m-2} \text{ merges two Gaussian} \quad (16)\\
&= ...\\
&= \sqrt{\prod_{i=1}^m \alpha^m}X^0 + \sqrt{1 - \prod_{i=1}^m \alpha^m}\overline{\epsilon}^0 \quad (17)\\
&= \sqrt{\overline{\alpha}^m}X^0 + \sqrt{1-\overline{\alpha}^m}\epsilon \text{ ;where } \epsilon = \overline{\epsilon}^0, \overline{\alpha}^m = \prod_{i=1}^m \alpha^m \quad (18)\\
&\sim \mathcal{N}(X^m; \sqrt{\overline{\alpha}^m}X^0, (1-\overline{\alpha}^m)\mathbf{I}) \quad (19)
\end{aligned}$$

The mergence from 15 to 16 is achieved by the additive property of a Gaussian distribution, *i.e.*,

$$\mathcal{N}(0, (\sigma_1^2 + \sigma_2^2)\mathbf{I}) = \mathcal{N}(0, \sigma_1^2\mathbf{I}) + \mathcal{N}(0, \sigma_2^2\mathbf{I}),$$

where $\sigma_1 = \sqrt{\alpha^m(1-\alpha^{m-1})}$ and $\sigma_2 = \sqrt{1-\alpha^m}$.

# E  Derivation of the Objective $\mathcal{L}_d$

Firstly, we derive how to obtain the $\mathcal{L}_{VLB}$ in Section 3.4.

$$\mathcal{L} \leq -\mathcal{L}_{VLB} = \mathbb{E}_q\Big[\log\frac{q(X_{1:M}|X^0)}{p_\Theta(X_{0:M})}\Big]$$

$$= \mathbb{E}_q\Big[\log\frac{\prod_{m=1}^M q(X^m|X^{m-1})}{p_\Theta(X^M|\mathbf{c}_\phi^M)\prod_{m=1}^M p_\Theta(X^{m-1}|X^m,\mathbf{c}_\phi^m)}\Big]$$

$$= \mathbb{E}_q\Big[-\log p_\Theta(X^M|\mathbf{c}_\phi^M) + \sum_{m=1}^M \log\frac{q(X^m|X^{m-1})}{p_\Theta(X^{m-1}|X^m,\mathbf{c}_\phi^m)}\Big]$$

$$= \mathbb{E}_q\Big[-\log p_\Theta(X^M|\mathbf{c}_\phi^M) + \sum_{m=2}^M \log\frac{q(X^m|X^{m-1})}{p_\Theta(X^{m-1}|X^m,\mathbf{c}_\phi^m)} + \log\frac{q(X^1|X^0)}{p_\Theta(X^0|X^1,\mathbf{c}_\phi^1)}\Big]$$

$$= \mathbb{E}_q\Big[-\log p_\Theta(X^M|\mathbf{c}_\phi^M) + \sum_{m=2}^M \log\frac{q(X^m|X^{m-1},X^0)}{p_\Theta(X^{m-1}|X^m,\mathbf{c}_\phi^m)} + \log\frac{q(X^1|X^0)}{p_\Theta(X^0|X^1,\mathbf{c}_\phi^1)}\Big]$$

$$= \mathbb{E}_q\Big[-\log p_\Theta(X^M|\mathbf{c}_\phi^M) + \sum_{m=2}^M \log\Big(\frac{q(X^{m-1}|X^m,X^0)}{p_\Theta(X^{m-1}|X^m,\mathbf{c}_\phi^m)}\cdot\frac{q(X^m|X^0)}{q(X^{m-1}|X^0)}\Big) + \log\frac{q(X^1|X^0)}{p_\Theta(X^0|X^1,\mathbf{c}_\phi^1)}\Big]$$

$$= \mathbb{E}_q\Big[-\log p_\Theta(X^M|\mathbf{c}_\phi^M) + \sum_{m=2}^M \log\frac{q(X^{m-1}|X^m,X^0)}{p_\Theta(X^{m-1}|X^m,\mathbf{c}_\phi^m)} + \sum_{m=2}^M \log\frac{q(X^m|X^0)}{q(X^{m-1}|X^0)} + \log\frac{q(X^1|X^0)}{p_\Theta(X^0|X^1,\mathbf{c}_\phi^1)}\Big]$$

$$= \mathbb{E}_q\Big[-\log p_\Theta(X^M|\mathbf{c}_\phi^M) + \sum_{m=2}^M \log\frac{q(X^{m-1}|X^m,X^0)}{p_\Theta(X^{m-1}|X^m,\mathbf{c}_\phi^m)} + \log\frac{q(X^m|X^0)}{q(X^1|X^0)} + \log\frac{q(X^1|X^0)}{p_\Theta(X^0|X^1,\mathbf{c}_\phi^1)}\Big]$$

$$= \mathbb{E}_q\Big[\log\frac{q(X^M|X^0)}{p_\Theta(X^M|\mathbf{c}_\phi^M)} + \sum_{m=2}^M \log\frac{q(X^{m-1}|X^m,X^0)}{p_\Theta(X^{m-1}|X^m,\mathbf{c}_\phi^m)} - \log p_\Theta(X^0|X^1,\mathbf{c}_\phi^1)\Big]$$

$$= \mathbb{E}_q\Big[\log\frac{q(X^M|X^0)}{p_\Theta(X^M|\mathbf{c}_\phi^M)}\Big] + \sum_{m=2}^M q(X^{m-1}|X^m,X^0)\log\frac{q(X^{m-1}|X^m,X^0)}{p_\Theta(X^{m-1}|X^m,\mathbf{c}_\phi^m)} - \log p_\Theta(X^0|X^1,\mathbf{c}_\phi^1)\Big]$$

$$= \mathbb{E}_q\Big[KL(q(X^M|X^0)\,\|\,p_\Theta(X^M|\mathbf{c}_\phi^M)) - \log p_\Theta(X^0|X^1,\mathbf{c}_\phi^1)$$

$$+ \sum_{m=2}^M KL(q(X^{m-1}|X^m,X^0)\,\|\,p_\Theta(X^{m-1}|X^m,\mathbf{c}_\phi^m))\Big].$$

We ignore the first term because both $q(X^M|X^0)$ and $p_\Theta(X^M|\mathbf{c}_\phi^M)$ are the same Gaussian distribution, so the KL divergence between them is equal to 0. Meanwhile, the second term can be formulated as the third term when $m=1$ because

$$-\log p_\Theta(X^0|X^1,\mathbf{c}_\phi^1) = \log\frac{1}{p_\Theta(X^0|X^1,\mathbf{c}_\phi^1)}$$

$$= q(X^0|X^1,X^0)\log\frac{q(X^0|X^1,X^0)}{p_\Theta(X^0|X^1,\mathbf{c}_\phi^1)}$$

$$= KL(q(X^0|X^1,X^0)\,\|\,p_\Theta(X^0|X^1,\mathbf{c}_\phi^1)).$$

Therefore, our aim is to calculate the third term. Let the mean and variance of $q(X^{m-1}|X^m,X^0)$ be $\widetilde{\mu}^m(X^m,X^0)$ and $\widetilde{\sigma}^m$, respectively. Let the mean and variance of $p_\Theta(X^{m-1}|X^m,\mathbf{c}_\phi^m)$ be $\mu_\Theta(X^m,m,\mathbf{c}_\phi^m)$ and $\widetilde{\sigma}^m$, respectively. We show the analytical solution for the KL divergence of two multivariate Gaussian distributions. Here, we omit its specific derivation and directly use the conclusion:

$$KL(\mathcal{N}(p;\mu_p,\Sigma_p)\,\|\,\mathcal{N}(q;\mu_q,\Sigma_q)) = \frac{1}{2}\Big[\log\frac{|\Sigma_q|}{|\Sigma_p|} - d + \text{tr}(\Sigma_q^{-1}\Sigma_p) + (\mu_q - \mu_p)^T\Sigma_q^{-1}(\mu_p - \mu_q)\Big],$$

$$(20)$$

where $p$ and $q$ represent two Gaussian distributions. $\mu$ amd $\Sigma$ are the mean vector and covariance matrix, respectively. $\text{tr}(\cdot)$ calculates the trace of a matrix. Thus, by substituting $q$ and $p_\Theta$ to Equation 20, we get,

$$
\begin{aligned}
&KL(q(X^{m-1}|X^m, X^0) \parallel p_\Theta(X^{m-1}|X^m, \mathbf{c}_\phi^m)) \\
=&KL(\mathcal{N}(X^{m-1}; \widetilde{\mu}^m(X^m, X^0), \widetilde{\sigma}^m\mathbf{I}) \parallel \mathcal{N}(X^{m-1}; \mu_\Theta(X^m, m, \mathbf{c}_\phi^m), \Sigma_q(m)\mathbf{I})) \\
=&\frac{1}{2}\Big[ \log\frac{|\widetilde{\sigma}^m\mathbf{I}|}{|\widetilde{\sigma}^m\mathbf{I}|} - d + \text{tr}((\widetilde{\sigma}^m\mathbf{I})^{-1}\widetilde{\sigma}^m\mathbf{I}) + (\mu_\Theta - \widetilde{\mu}^m)^T(\widetilde{\sigma}^m\mathbf{I})^{-1}(\mu_\Theta - \widetilde{\mu}^m)\Big]. \\
=&\frac{1}{2}[\log 1 - d + d + (\mu_\Theta - \widetilde{\mu}^m)^T(\widetilde{\sigma}^m\mathbf{I})^{-1}(\mu_\Theta - \widetilde{\mu}^m)] \\
=&\frac{1}{2}[(\mu_\Theta - \widetilde{\mu}^m)^T(\widetilde{\sigma}^m\mathbf{I})^{-1}(\mu_\Theta - \widetilde{\mu}^m)] \\
=&\frac{1}{2\widetilde{\sigma}^m\mathbf{I}} \parallel \widetilde{\mu}^m - \mu_\Theta \parallel_2^2 \\
\propto& \parallel \widetilde{\mu}^m(X^m, X^0) - \mu_\Theta(X^m, m, \mathbf{c}_\phi^m) \parallel_2^2)
\end{aligned}
\tag{21}
$$

We can see that the objective is transformed to minimize the $l2$ distance between mean of true posterior distribution and estimated distribution. This equation can be further simplified by substituting specific expressions of $\widetilde{\mu}^m(X^m, X^0)$ and $\mu_\Theta(X^m, m, \mathbf{c}_\phi^m)$ to it. To calculate $\widetilde{\mu}^m(X^m, X^0)$, we use the Bayes formula:

$$
q(X^{m-1}|X^m, X^0) = q(X^m|X^{m-1}, X^0)\frac{q(X^{m-1}|X^0)}{q(X^m|X^0)} = q(X^m|X^{m-1})\frac{q(X^{m-1}|X^0)}{q(X^m|X^0)}.
\tag{22}
$$

After applying the Bayes formula, we calculate each term using Equation 19 and substitute them to the above Equation. Then, we have

$$
\begin{aligned}
&q(X^{m-1}|X^m, X^0) = q(X^m|X^{m-1})\frac{q(X^{m-1}|X^0)}{q(X^m|X^0)} \\
\propto& \exp\Big(-\frac{1}{2}\Big(\frac{(X^m - \sqrt{\alpha^m}X^{m-1})^2}{\beta^m} + \frac{(X^{m-1} - \sqrt{\bar{\alpha}^{m-1}}X^0)^2}{1 - \bar{\alpha}^{m-1}} - \frac{(X^m - \sqrt{\bar{\alpha}^m}X^0)^2}{1 - \bar{\alpha}^m}\Big)\Big) \\
=& \exp\Big(-\frac{1}{2}\Big(\frac{(X^m)^2 - 2\sqrt{\alpha^m}X^mX^{m-1} + \alpha^m(X^{m-1})^2}{\beta^m} \\
&+ \frac{(X^{m-1})^2 - 2\sqrt{\bar{\alpha}^{m-1}}X^0X^{m-1} + \bar{\alpha}^{m-1}(X^0)^2}{1 - \bar{\alpha}^{m-1}} - \frac{(X^m - \sqrt{\bar{\alpha}^m}X^0)^2}{1 - \bar{\alpha}^m}\Big)\Big) \\
=& \exp\Big(-\frac{1}{2}\Big(\underbrace{(\frac{\alpha^m}{\beta^m} + \frac{1}{1 - \bar{\alpha}^{m-1}})(X^{m-1})^2}_{\text{Variance term of } X^{m-1}} - \underbrace{(\frac{2\sqrt{\alpha^m}}{\beta^m}X^m + \frac{2\sqrt{\bar{\alpha}^{m-1}}}{1 - \bar{\alpha}^{m-1}}X^0)X^{m-1}}_{\text{Mean term of } X^{m-1}} + \underbrace{C(X^m, X^0)}_{\text{Term unrelated to } X^{m-1}}\Big)\Big)
\end{aligned}
\tag{23}
$$

The exponential part of the probability density function of a Gaussian distribution can be expressed as:

$$
\exp\Big(-\frac{(X - \mu)^2}{2\sigma^2}\Big) = \exp\Big(-\frac{1}{2}(\frac{1}{\sigma^2}X^2 - \frac{2\mu}{\sigma^2}X + \frac{\mu^2}{\sigma^2})\Big)
\tag{24}
$$

By comparing Equation 24 and 23, we obtain the mean $\widetilde{\mu}^m(X^m, X^0)$ and variance $\widetilde{\beta}^m$ of the Gaussian distribution $q(X^{m-1}|X^m, X^0)$,

$$\tilde{\beta}^m = \sigma^2 = 1/(\frac{\alpha^m}{\beta^m} + \frac{1}{1 - \bar{\alpha}^{m-1}}) = 1/(\frac{\alpha^m - \bar{\alpha}^m + \beta^m}{\beta^m(1 - \bar{\alpha}^{m-1})}) = \frac{1 - \bar{\alpha}^{m-1}}{1 - \bar{\alpha}^m} \cdot \beta^m \quad (25)$$

$$\tilde{\mu}^m(X^m, X_0) = (\frac{\sqrt{\alpha^m}}{\beta^m} X^m + \frac{\sqrt{\bar{\alpha}^{m-1}}}{1 - \bar{\alpha}^{m-1}} X_0)/(\frac{\alpha^m}{\beta^m} + \frac{1}{1 - \bar{\alpha}^{m-1}})$$

$$= (\frac{\sqrt{\alpha^m}}{\beta^m} X^m + \frac{\sqrt{\bar{\alpha}^{m-1}}}{1 - \bar{\alpha}^{m-1}} X_0)\frac{1 - \bar{\alpha}^{m-1}}{1 - \bar{\alpha}^m} \cdot \beta^m$$

$$= \frac{\sqrt{\alpha^m}(1 - \bar{\alpha}^{m-1})}{1 - \bar{\alpha}^m} X^m + \frac{\sqrt{\bar{\alpha}^{m-1}}\beta^m}{1 - \bar{\alpha}^m} X_0 \quad (26)$$

Further, we calculate $X^0$ using Equation 18 and substitute it to Equation 26 to get the expression related to $\epsilon$,

$$X^0 = \frac{1}{\sqrt{\bar{\alpha}^m}}(X^m - \sqrt{1 - \bar{\alpha}^m}\epsilon) \quad (27)$$

$$\tilde{\mu}^m(X^m, X_0) = \frac{\sqrt{\alpha^m}(1 - \bar{\alpha}^{m-1})}{1 - \bar{\alpha}^m} X^m + \frac{\sqrt{\bar{\alpha}^{m-1}}\beta^m}{1 - \bar{\alpha}^m} \cdot \frac{1}{\sqrt{\bar{\alpha}^m}}(X^m - \sqrt{1 - \bar{\alpha}^m}\epsilon)$$

$$= \frac{1}{\sqrt{\alpha^m}}(X^m - \frac{\beta^m}{\sqrt{1 - \bar{\alpha}^m}}\epsilon) \quad (28)$$

By replacing $\epsilon$ with $\epsilon_\Theta(X^m, m, \mathbf{c}_\phi^m)$, we obtain the expression of $\mu_\Theta(X^m, m, \mathbf{c}_\phi^m)$,

$$\mu_\Theta(X^m, m, \mathbf{c}_\phi^m) = \frac{1}{\sqrt{\alpha^m}}(X^m - \frac{\beta^m}{\sqrt{1 - \bar{\alpha}^m}}\epsilon_\Theta(X^m, m, \mathbf{c}_\phi^m)) \quad (29)$$

We substitute Equation 26 and 29 to Equation 21, and the simplified objective can be represented as,

$$\mathcal{L}_{unobs} = \mathbb{E}_{X_{unobs}^0, \epsilon, m}||\epsilon - \epsilon_\Theta(X_{unobs}^m, m, \mathbf{c}_{\phi,unobs}^m)||_2$$

$$\mathcal{L}_{fut} = \mathbb{E}_{X_{fut}^0, \epsilon, m}||\epsilon - \epsilon_\Theta(X_{fut}^m, m, \mathbf{c}_{\phi,fut}^m)||_2$$

$$\mathcal{L}_d = \mathcal{L}_{fut} + \mathcal{L}_{unobs} \quad (30)$$

# F   Visualization of Hard Cases

We provide more hard cases in Figure 4, we observe our method can better handle the hard cases and predict more accurate trajectories than other methods.

# G   Visualization of Failure Cases

As shown in Figure 5, our model sometimes fails when pedestrians start to walk or abruptly change the intention. We will further optimize our method according to these failure cases in our future work.