# OpenReview forum: "BCDiff: Bidirectional Consistent Diffusion for Instantaneous Trajectory Prediction"
_NeurIPS.cc/2023/Conference — NeurIPS 2023 poster_

### Official Review · Reviewer_JXan · 2023-07-01

**Soundness:** 3 good
**Presentation:** 3 good
**Contribution:** 2 fair
**Rating:** 6
**Confidence:** 4

**Summary:**

This paper proposed a  diffusion-based model specifically for instantaneous trajectory prediction. It leverages the consistency of the prediction for unobserved historical trajectory and the future trajectory to improve the accuracy of the future trajectory. Regarding the noise in the prediction wh, especially in the early steps, the paper introduced a gating mechanism to guide the fusion.

**Strengths:**

1. The paper is written-well and the method section is generally straightforward and coherent.
2. The idea is intuitive. Since longer trajectories would provide more dynamic information and motion hints to help the system make better predictions, even those longer trajecotires are also predicted and contain a certain amount of noise.
3. The experiments and comparison are thorough and have covered two popular datasets in pedestrian trajectory forecasting and three SOTA methods.
4. The improvements are significant compared to the existing instantaneous trajectory forecasting methods.

**Weaknesses:**

Maybe there is some misunderstanding but my major concern is that the motivation of the method may not align with the motivation of the task very well.

The task of instantaneous trajectory prediction wants the system to extract valuable information when the objects' dynamic information is extremely limited.  As the authors mentioned in the related works L110-L111, the lack of temporal information is an inherent challenge of the task. Therefore, MOE[1] tries to use map information as an informative reference to help the system conduct more accurate predictions, and the motivation of DTO[2] is to do a thorough analysis of how limited observed information can lead to comparatively accurate predictions.

But in this paper, the authors let the system access the unobserved historical trajectories during the training time as one of the supervision signals. This data-augmentation-like design may make the comparison with other methods not fair enough. It would be necessary to rule out the influence of data augmentation when evaluating the boost brought by the system and prove the performance boost all comes from that the system has fully dug the potential of the limited observation.

[1] Jianhua Sun, Yuxuan Li, Liang Chai, Hao-Shu Fang, Yong-Lu Li, and Cewu Lu. Human trajectory prediction with momentary observation. In Proceedings of the IEEE/CVF Conference on Computer Vision and Pattern Recognition, pages 6467–6476, 2022.

[2] Alessio Monti, Angelo Porrello, Simone Calderara, Pasquale Coscia, Lamberto Ballan, and Rita Cucchiara. How many observations are enough? knowledge distillation for trajectory forecasting. In Proceedings of the IEEE/CVF Conference on Computer Vision and Pattern Recognition, pages 6553–6562, 2022.





**Questions:**

1. Given the first point in [Weakness], for a fair comparison, I was curious to know the evaluation results if the existing methods also utilize unobserved trajectories as a prediction target for data augmentation. For instance, in Table 1, during the training of Trajectron++, Trajectron++ with MOE, and Trajectron++ with DTO, in addition to supervising future trajectories, we reverse the observed historical trajectories, use them as input to predict unobserved trajectories and evaluate the models. Another approach to demonstrate this could be seen in Table 3, BCDiff-w/o.Guidance, where instead of employing two diffusion models, only one model is used to predict both reversed unobserved and future trajectories.

2. Regarding the gating mechanism, it is intuitive that the early steps should have lower weights given the noise level. Have authors tried simpler ways to handle the issue? Such as manually assigning lower weights to the early steps, and gradually increasing the weight till the last steps.

3. Since the behavior of pedestrians can be very spontaneous, can you provide the evaluation comparison with other methods on some hard cases, especially for instantaneous trajectory prediction? Such as the cases where there are large contrasts between the unobserved trajectories and the future trajectories.

4. Table 4 is very interesting. Can you provide some more quantitative results related to it? For instance, the numeric results of the accuracy on unobserved trajecotires and the relationship between which and the performance of the future predictions.

**Limitations:**

The authors have not discussed the limitation. From my point of view, the limitation of the work is that the input information is so little that it can hardly provide valid predictions for samples in the wild or more complex situations. Furthermore, the dynamic or interaction types of the tested datasets are also very limited, the results can be biased without testing the methods on more diverse datasets, such as some autonomous driving datasets.

---

> ### Author Rebuttal · Authors · 2023-08-09
>
> >Q1: The authors let the system access the unobserved historical trajectories during the training time as one of the supervision signals. It would be necessary to rule out the influence of unobserved historical trajectories when evaluating the boost brought by the system and prove the performance boost all comes from that the system has fully dug the potential of the limited observation.
>
> A1: Thanks for your suggestions for improving the quality of our method. Following your suggestions, we modify the baseline Trajectron++, Trajectron++ (MOE), and Trajectron++ (DTO) with the supervision of both future trajectories and unobserved trajectory ground-truth. The results are listed below. We observe our BCDiff still outperforms existing methods, which rules out the influence of unobserved trajectories and demonstrates the superiority of the proposed approach.
>
> |Model|Methods|||Dataset||||
> |-|-|-|-|-|-|-|-|
> |||ETH|Hotel|UNIV|ZARA1|ZARA2|AVG|
> ||Instantaneous|0.71/1.36|0.25/0.50|0.34/0.74|0.22/0.42|0.18/0.34|0.34/0.67|
> ||MOE|0.63/1.10|0.18/0.31|0.32/0.59|0.22/0.41|0.17/0.32|0.30/0.55|
> |Trajectron++|MOE w/o Image|0.66/1.16|0.22/0.39|0.33/0.62|0.22/0.42|0.18/0.33|0.32/0.58|
> ||DTO|0.70/1.21|0.22/0.45|0.31/0.60|0.22/0.41|0.17/0.33|0.32/0.60|
> ||BCDiff|0.61/1.09|0.16/0.28|0.28/0.53|0.22/0.41|0.18/0.33|0.29/0.53|
>
> >Q2: Regarding the gating mechanism, Have authors tried simpler ways to assign weight?
>
> A2: Thanks for your suggestions. We add an experiment to adjust the weights manually. We linearly increase the weights from 0 to 1 with a step size of $1/M$, where $M$ is the number of steps in the diffusion model. The results of using heuristic weight assigning (BCDiff-manually) are indeed better than that of using equal weights (BCDiff-w/oGate). Importantly, our gate mechanism achieves the best performance, since our method is expected to automatically learn the optimal weights through gradient descent.
>
> |Method|ETH/UCY|
> |-|-|
> |BCDiff-w/o Gate|0.32/0.57|
> |BCDiff-manually|0.31/0.55|
> |BCDiff-Gate|0.29/0.53|
>
>
> >Q3: Can you provide comparisons with other methods on some hard cases for instantaneous trajectory prediction?
>
> A3: Thanks for your suggestions. We provide some hard cases in the attached PDF file in the Global Response. As shown in Figure 1, we observe our method can better handle the hard cases and predict more accurate trajectories than other methods.
>
> >Q4: Can you provide more quantitative results related to Table 4?
>
> A4: Thanks for your suggestions. We provide the ADE/FDE of unobserved trajectories, as shown in the below table.
>
> As the number of unobserved points ($T_{unobs}$) increases, the accuracy of unobserved trajectory prediction decreases. This is reasonable since it becomes more difficult when more points need to be predicted. In addition, as $T_{unobs}$ increases, the accuracy of future trajectory prediction gradually improves. This can be attributed to leveraging the additional unobserved points to provide useful information.  When $T = 5$, the prediction performance starts to decline. This is because it becomes imprecise when predicting previous unobserved historical points with a larger length, thus introducing noise into our method.
>
> |Dataset|Type|$T_{unobs}=1$|$T_{unobs}=2$|$T_{unobs}=3$|$T_{unobs}=4$|$T_{unobs}=5$|$T_{unobs}=6$|
> |-|-|-|-|-|-|-|-|
> |ETH/UCY|Future|0.31/0.55|0.30/0.54|0.30/0.53|**0.29/0.53**|0.32/0.57|0.34/0.60|
> ||Unobserved|0.006/0.006|0.034/0.033|0.055/0.063|0.086/0.089|0.108/0.133|0.149/0.212|
> |SDD|Future|11.86/19.78|11.81/19.71|11.68/19.50|**11.56/19.32**|11.92/20.01|12.33/20.98|
> ||Unobserved|0.311/0.312|1.564/1.492|2.626/2.678|3.552/3.884|4.828/5.423|6.177/7.894|
>
> >Q5: Limitation of the work: Not enough input information for accurate predictions in complex situations. Tested datasets have limited types of movement and interaction, which could make results biased. Need to test on diverse datasets.
>
> A5: Thanks for your comments.
> We put the limitations in Section B of the Appendix due to space limitations.
> Pedestrian trajectory prediction is an important research problem for autonomous driving [1,2], and many works have been proposed for studying it in the past decade [3,4]. Thus, we mainly focus on the pedestrian trajectory prediction problem in this work.
> Different from most of the existing pedestrian trajectory prediction methods, we consider a more challenging scenario, where merely two frames of trajectories can be observed to predict future trajectories.
>
> It is indeed an interesting idea to apply our method to predict trajectories on autonomous driving datasets [5,6] for other moving agents, e.g., vehicles. We are currently considering such a problem setting and look forward to experimenting with our method. It is anticipated for our method to work well on vehicles. The reasons are two-fold:
>
> 1. Pedestrian trajectory prediction is a more challenging problem, due to the moving directions of pedestrians being more random than those of vehicles driving on the road.
>
> 2. Our proposed BCDiff is an encoder-agnostic framework that operates independently of specific encoders. It can be integrated with diverse existing trajectory prediction methods. By leveraging a strong encoder, intricate scenarios, and interactions can be fully captured and represented. Moreover, we will be also actively committed to further explore for effectively addressing complex environmental dynamics in our approach, thereby enhancing the accuracy of predictive capabilities.
>
> Reference
>
> [1] Codes for View Vertically: A Hierarchical Network for Trajectory Prediction via Fourier Spectrums. ECCV'22
>
> [2] SocialVAE: Human Trajectory Prediction using Timewise Latents. ECCV'22
>
> [3] Human Trajectory Prediction via Neural Social Physics. ECCV'22
>
> [4] Human Trajectory Prediction with Momentary Observation. CVPR'22
>
> [5] Argoverse: 3D Tracking and Forecasting with Rich Maps. CVPR'19
>
> [6] Large Scale Interactive Motion Forecasting for Autonomous Driving: The Waymo Open Motion Dataset. ICCV'21

---

### Official Review · Reviewer_WQWs · 2023-07-04

**Soundness:** 4 excellent
**Presentation:** 4 excellent
**Contribution:** 3 good
**Rating:** 7
**Confidence:** 4

**Summary:**

In this paper, the authors propose a diffusion framework for instantaneous trajectory prediction (only observing two frames in history). Specifically, they develop two coupled diffusion models by designing a mutual guidance mechanism that can bidirectionally and consistently generate unobserved historical trajectories and future trajectories step-by-step, to utilize the complementary information between them. The experimental results demonstrate the proposed solution improves the accuracy of instantaneous trajectory prediction.

**Strengths:**

1. The intuition of the proposed solution is compelling. As mentioned in line 60, "Both previous unobserved historical trajectories and future trajectories contain information of the same pedestrian at different timesteps, and thereby they provide complementary information to each other. It will be beneficial for the prediction of future trajectories if we can design an elegant method to simultaneously generate previous unobserved historical trajectories and future trajectories by fully leveraging the complementary information between them."
2. The proposed solution (that simultaneously generates both future and unobserved historical trajectories in a consistent manner) sounds solid and should work in theory.
3. The experimental results on multiple datasets and models prove the validity of the proposed solution.
4. This paper is well-organized and easy to understand.
5. The supplementary material provides more details about the model and experiments, which makes it easier for the reader to understand their work.




**Weaknesses:**

1. The proposed solution is time-consuming (4 seconds for predicting trajectories once). This problem appears to be particularly serious, especially in the area of autonomous driving. The long computation time should be caused by the model design (diffusion model and using an iterative way to generate predicted trajectories).
2. Some details are missing. For example, what are the social and scene context e, and how to generate them for the Encoder is not clear.
3. It will be better to show some failure cases. Considering that the trajectory prediction task is usually used in autonomous driving, failure cases are more valuable to study.

**Questions:**

1. Providing more information about failure cases will help readers better understand the performance of the proposed solution.
2. Discussing some potential ways to speed up the proposed solution will be better.


**Limitations:**

The long computation time should be the biggest limitation of the proposed solution. The authors discussed the issues in the supplementary material, but no potential doable solution. It may need more exploration.

---

> ### Author Rebuttal · Authors · 2023-08-09
>
> >Q1: The proposed solution is time-consuming (4 seconds for predicting trajectories once). This problem appears to be particularly serious, especially in the area of autonomous driving. The long computation time should be caused by the model design (diffusion model and using an iterative way to generate predicted trajectories). Discussing some potential ways to speed up the proposed solution will be better.
>
> A1: Thanks for your comments. We admit the inference time is one of our limitations due to the nature of the diffusion model, and have put it as a limitation in Appendix B of Supplementary Material.
> Currently, we primarily focus on the accuracy of instantaneous trajectory predictions. We will further consider how to accelerate it in our future work. The following are some potentially feasible acceleration strategies:
>
> 1. From the perspective of diffusion algorithms: Our current framework is built based on a typical diffusion model, DDPM. DDPM models the trajectory generation process as a Markov process that requires continuous M-steps to generate trajectories. There are many cutting-edged methods for accelerating inference [1,2,3,4], by relaxing the constraints of the Markov process and generating trajectories using skipping steps. In this way, the model can use less steps to generate a trajectory, and thereby shorten the reference time.
>
> 2. From the perspective of model structure:} We can combine our method with Half-Precision/Model Quantization techniques [7,8] to lower the computational cost, or Model Pruning [5,6] to simplify the model, or Knowledge Distillation [9, 10] to utilize a small but effective model for inference.
>
> We will add them in the final version.
>
> >Q2: Some details are missing. For example, what are the social and scene context e, and how to generate them for the Encoder is not clear.
>
> A2: Sorry for confusing you. The social context is the information of other moving pedestrians around the target pedestrian whose future trajectories are to be predicted. The scene context represents the map information around the target agent. Generally speaking, these two kinds of information will be integrated into the encoder of the trajectory prediction model to boost the representation ability of historical trajectories. Since our proposed BCDiff is actually a decoder, in other words, it is an encoder-agnostic framework, our attention is not paid to the contextual information. We will include the above details in the final version.
>
> >Q3: It will be better to show some failure cases. Considering that the trajectory prediction task is usually used in autonomous driving, failure cases are more valuable to study.
>
> A3: Thanks for your suggestions. We provide some failure cases in the attached PDF file in the Global Response. As shown in Figure 2, our model sometimes fails when pedestrians start to walk or abruptly change the intention. We will further optimize our method according to these failure cases in our future work.
>
> Reference
>
> [1] DPM-Solver: A Fast ODE Solver for Diffusion Probabilistic Model Sampling in Around 10 Steps. NeurIPS'22
>
> [2] Leapfrog Diffusion Model for Stochastic Trajectory Prediction. CVPR'23
>
> [3] Progressive Distillation for Fast Sampling of Diffusion Models. ICLR'22
>
> [4] Denoising Diffusion Implicit Models. ICLR'21
>
> [5] Structural Pruning for Diffusion Models. arXiv'23
>
> [6] Prospect Pruning: Finding Trainable Weights at Initialization using Meta-Gradients. ICLR'22
>
> [7] Q-Diffusion: Quantizing Diffusion Models. arXiv'23
>
> [8] Post-Training Quantization on Diffusion Models. CVPR'23
>
> [9] On Distillation of Guided Diffusion Models. CVPR'23
>
> [10] Progressive Distillation for Fast Sampling of Diffusion Models. ICLR'22

---

> > ### Comment · Reviewer_WQWs · 2023-08-12
> >
> > Thank you so much for your rebuttal. It answers my questions, and the authors also have plans for future work. I do not have further questions and will maintain my positive comment for this paper.

---

### Official Review · Reviewer_VvSx · 2023-07-05

**Soundness:** 2 fair
**Presentation:** 3 good
**Contribution:** 3 good
**Rating:** 6
**Confidence:** 3

**Summary:**

This paper focuses on the trajectory prediction in an instantaneous observation by generative diffusion models. Specifically, the authors propose a bidirectional consistent diffusion framework, named BCDiff, to address the issue of temporal information scarcity. BCDiff employs two coupled diffusion models are devised to predict unobserved historical trajectories and future trajectories with a mutual guidance mechanism, which ensures the consistency between predicted unobserved trajectories and predicted future trajectories.  The authors conduct extensive experiments to show the effectiveness of proposed method.

**Strengths:**

Instantaneous trajectory prediction is a common challenge in traffic scene. It is promising that employ the diffusion model in instantaneous trajectory prediction.  The conducted experimental results show proposed BCDiff is effective in the scenes of ETH-UCY and SDD.
Moreover, the authors claim that the proposed method has friend compatibility with existing trajectory prediction models. Also, the organization of paper is well.

**Weaknesses:**

The experiments is insufficient.
1: The compared baselines are not the newest methods.  The authors should add more baselines with high accuracy, such as SocialVAE[1], SIT[2], MemoNet[3]. These baselines could be trained under the setting of the instantaneous trajectory prediction, such as observing 2 steps and predicting the next 12 steps.
2: The inference time is missing. Instantaneous trajectory prediction often occurs on the emergency scene. The inference time is an important metrics to evaluate model.

[1] Pei Xu, Jean-Bernard Hayet, and Ioannis Karamouzas. Socialvae: Human trajectory prediction using timewise latents. In ECCV, pages 511–528. Springer, 2022
[2] Liushuai Shi, Le Wang, Chengjiang Long, Sanping Zhou, Fang Zheng, Nanning Zheng, and Gang Hua. Social interpretable tree for pedestrian trajectory prediction. In AAAI, pages 2235–2243, 2022
[3] Chenxin Xu, Weibo Mao, Wenjun Zhang, and Siheng Chen. Remember intentions: Retrospective-memory-based trajectory prediction. In CVPR, pages 6488–6497, 2022.

**Questions:**

As shown in WEAKNESS,
1)	I suggest the authors provide more comparison with the latest baselines in the setting of instantaneous trajectory prediction.
2)	Standard diffusion model suffer from expensive inference time, the author should provide detailed inference speed to evaluate the proposed model comprehensively.


**Limitations:**

Refer to the weakness and questions. My concerns mainly lies on the comparison with the latest baselines and the inference time.  Providing above information could be helpful to improve this paper comprehensively.

---

> ### Author Rebuttal · Authors · 2023-08-09
>
> >Q1: The compared baselines are not the newest methods. The authors should add more baselines with high accuracy, such as SocialVAE, SIT, MemoNet. These baselines could be trained under the setting of the instantaneous trajectory prediction, such as observing 2 steps and predicting the next 12 steps.
>
> A1: Following your suggestion, we add experiments to compare with SocialVAE [3] that is proposed recently on ETH/UCY Dataset.
> The results are listed in the below table. We observe that our BCDiff still performs better than baselines, which demonstrates the superiority of our proposed method for instantaneous trajectory prediction. We will include the results in the final version.
>
> | Model        | Methods       |           |           | Dataset   |           |           |           |
> |--------------|---------------|-----------|-----------|-----------|-----------|-----------|-----------|
> |              |               | ETH       | Hotel     | UNIV      | ZARA1     | ZARA2     | AVG       |
> |              | Instantaneous | 0.64/1.10 | 0.21/0.34 | 0.27/0.51 | 0.22/0.39 | 0.18/0.34 | 0.30/0.54 |
> |              | MOE [1]       | 0.57/1.01 | 0.17/0.29 | 0.26/0.44 | 0.22/0.36 | 0.17/0.32 | 0.28/0.48 |
> | SocialVAE[3] | MOE w/o Image | 0.59/1.06 | 0.20/0.33 | 0.27/0.49 | 0.22/0.38 | 0.18/0.33 | 0.29/0.52 |
> |              | DTO [2]       | 0.61/1.06 | 0.18/0.31 | 0.25/0.43 | 0.22/0.38 | 0.17/0.33 | 0.29/0.50 |
> |              | BCDiff        | 0.53/0.91 | 0.17/0.27 | 0.24/0.40 | 0.21/0.37 | 0.16/0.26 | 0.26/0.44 |
>
> >Q2: The inference time is missing. Instantaneous trajectory prediction often occurs on the emergency scene. The inference time is an important metric to evaluate the model.
>
> A2: Thanks for your comments. We admit that inference time is one of our limitations due to the nature of the diffusion model, and we have put it as a limitation in the Section B of Supplementary Material.
> Currently, we primarily focus on the accuracy of instantaneous trajectory predictions. We will further consider how to accelerate it in our future work. The following are some potentially feasible acceleration strategies:
>
> 1. From the perspective of diffusion algorithms: Our current framework is built based on a typical diffusion model, DDPM. DDPM models the trajectory generation process as a Markov process that requires continuous M-steps to generate trajectories. There are many cutting-edge methods for accelerating inference [4,5,6,7], by relaxing the constraints of the Markov process and generating trajectories using skipping steps. In this way, the model can use fewer steps to generate a trajectory, and thereby shorten the reference time.
>
> 2. From the perspective of model structure: We can combine our method with Half-Precision/Model Quantization techniques [8,9] to lower the computational cost, Model Pruning [10,11] to simplify the model or Knowledge Distillation [12,13] to utilize a small but effective model for inference.
>
> Reference
>
> [1] Human Trajectory Prediction with Momentary Observation. CVPR'22
>
> [2] How Many Observations Are Enough? Knowledge Distillation for Trajectory Forecasting. CVPR'22
>
> [3] SocialVAE: Human Trajectory Prediction using Timewise Latents. ECCV'22
>
> [4] DPM-Solver: A Fast ODE Solver for Diffusion Probabilistic Model Sampling in Around 10 steps. NeurIPS'22
>
> [5] Leapfrog Diffusion Model for Stochastic Trajectory Prediction. CVPR'23
>
> [6] Progressive Distillation for Fast Sampling of Diffusion Models. ICLR'22
>
> [7] Denoising Diffusion Implicit Models. ICLR'21
>
> [8] Structural Pruning for Diffusion Models. arXiv'23
>
> [9] Prospect Pruning: Finding Trainable Weights at Initialization using Meta-Gradients. ICLR'22
>
> [10] Q-Diffusion: Quantizing Diffusion Models. arXiv'23
>
> [11] Post-Training Quantization on Diffusion Models. CVPR'23
>
> [12] On Distillation of Guided Diffusion Models. CVPR'23
>
> [13] Progressive Distillation for Fast Sampling of Diffusion Models. ICLR'22

---

### Official Review · Reviewer_ytbj · 2023-07-06

**Soundness:** 3 good
**Presentation:** 3 good
**Contribution:** 3 good
**Rating:** 6
**Confidence:** 3

**Summary:**

The paper presents a bi-directional consistent diffusion approach for handling instantaneous trajectory prediction, i.e. trajectory prediction without requiring a large number of observation time. It relies on two coupled diffusion models to generate historical and future trajectories. The forward diffusion model predicts future trajectories conditioned on the predicted trajectories of the other model and the limited observed trajectories, while the backward model predicts the historical trajectories guided by the predicted future trajectories and the observed trajectories. Since the amount of noise in the first few diffusion steps is inherently relatively high, a gating mechanism is introduced to control the contribution of the two types of trajectories. In the evaluation, the approach achieves state-of-the-art performance on the TH/UCY and Stanford Drone datasets by replacing the decoder of three popular trajectory prediction models with the proposed one.



Having read the reviews and the rebuttal, I am going to maintain my positive assessment. The rebuttal addressed a lot of concerns, especially the experimental ones.

**Strengths:**

- The idea and method of the paper are easy to follow.

- The use of diffusion models for trajectory generation is novel and interesting.

- The approach works even with very sparse observations based on the experimental evaluation.

- There are extensive experiments with different backbone architectures and datasets.

- The proposed approach beats the current state-of-the-art in almost every experiment.

**Weaknesses:**

- The task of instantaneous trajectory prediction is not standard. It would value for the approach to be evaluated for the standard trajectory prediction protocol. Then it would be allowed to be compared with a wide range of methods. This is a particularly interesting point since the diffusion models are heavy in terms of parameters. It would allow for a comparison with the current state-of-the-art methods.

- It is not clear how many observations are being used during the test time, for example, if it is $M = 7$, then the approach does not seem to improve the current state-of-the-art in trajectory. prediction. Moreover, it would be interesting to have an ablation study on the influence of step M on performance.

**Questions:**

- Clarify on the possibility to perform standard trajectory prediction and also discus the hyper-parameter selection.

**Limitations:**

Not applicable.

---

> ### Author Rebuttal · Authors · 2023-08-09
>
> >Q1: The task of instantaneous trajectory prediction is not standard. It would value for the approach to be evaluated for the standard trajectory prediction protocol. Then it would be allowed to be compared with a wide range of methods. This is a particularly interesting point since the diffusion models are heavy in terms of parameters. It would allow for a comparison with the current state-of-the-art methods.
>
> A1: Thanks for your suggestions. We would like to kindly remind you that there might be a misunderstanding regarding our paper caused by the problem formulation. The standard trajectory prediction takes 8 frames as observations to predict future trajectories. Instead, we focus on utilizing merely 2 frames of observations to predict future trajectories, which strictly follows the pioneering work MOE [1] for instantaneous trajectory prediction. Therefore, due to the different settings in terms of the number of observation points, it is unfair to compare instantaneous trajectory prediction methods with standard trajectory prediction methods. We will refine the problem formulation to be clearer in the final version.
>
> >Q2: It is not clear how many observations are being used during the test time, for example, if it is $M=7$, then the approach does not seem to improve the current state-of-the-art in trajectory prediction. Moreover, it would be interesting to have an ablation study on the influence of step $M$ on performance.
>
> A2: We apologize again for confusing you, and we'd like to kindly remind you that the notation $M$ you mentioned in the Question denotes the maximum diffusion steps in the diffusion mode, instead of the number of observation points.
>
> Then, as we mentioned in the A1 that we focus on the instantaneous trajectory prediction, where 2 frames of observations are input, studying trajectory prediction with the arbitrary number of observation points as input is beyond the research scope of this paper. We are instead committed to improving accuracy in the instantaneous setting with only two observation points.
>
> However, as you suggested, using the arbitrary lengths of observed trajectories to predict future trajectories is interesting to study. We are willing to explore the topic in future works.
>
>
> Reference
>
> [1] Human Trajectory Prediction with Momentary Observation. CVPR'22

---

> > ### Comment · Reviewer_ytbj · 2023-08-12
> >
> > Having read the reviews and the rebuttal, I am going to maintain my positive assessment. The rebuttal addressed a lot of concerns, especially the experimental ones.

---

### Official Review · Reviewer_GWuJ · 2023-07-09

**Soundness:** 2 fair
**Presentation:** 3 good
**Contribution:** 2 fair
**Rating:** 6
**Confidence:** 5

**Summary:**

The authors propose a bidirectional consistent diffusion framework for instantaneous trajectory prediction. Specifically, two coupled diffusion models are employed to predict unobserved historical and future trajectories jointly. In addition, a gating mechanism is presented to deal with the noisy situation at the initial steps. Experiments on ETH-UCY and SDD show that the proposed method is able to achieve the promising performance.

**Strengths:**

1. Clear-written and organized.
2. Promising results on several datasets.

**Weaknesses:**

1. I am concerned about the consistency between the discussed motivation and the experiments. In the Abstract and Introduction Section, the motivation is that "there are many real-world situations where the model lacks sufficient time to observe". However, in the Experiments Section, the evaluation protocol is the commonly used leave-one-out setting, which is somewhat inconsistent with motivation. I assume the training and the inference phase both take only 2 frames as the input (this is another unclear point), which makes the  "real-world situation" that the training scenario lacks adequate observation and the testing scenario also lacks adequate observation.

2. In addition to the above point, another concern is that only human trajectories are predicted. While such real-world situations often happen in the driving scenario.

3. To better demonstrate the motivation, some motivated experiments could be included, i.e., existing methods fail to make accurate predictions when the observations are not adequate.

4. In Lines 60-62, it claims that jointly predicting unobserved historical trajectories and future trajectories can be beneficial to each other. This claim is very similar to [1]. However, there is no experiment on this point, such as conducting these two jobs separately, or first predicting unobserved historical trajectories and then predicting future trajectories based on the predicted historical results.

5. In the Method Section, LSTM is used as in Eq.5, why don't use the Bi-directional LSTM?

6. Some details are missing. In my opinion, there is a gap between the training and the inference phase. During the training, the method has access to the future gound-truth trajectory, in other words, there could have a forward and backward process. While in the inference phase, the method mustn't have access to future information, and the details of how to evaluate the trained model are missing.



[1] Uncovering the Missing Pattern: Unified Framework Towards Trajectory Imputation and Prediction. CVPR'23

**Questions:**

My questions are raised in the above Weaknesses Section.

**Limitations:**

I didn't see any limitations discussed in the submission.

---

> ### Author Rebuttal · Authors · 2023-08-04
>
> >Q1: I am concerned about the consistency between the motivation and experimental setting.
>
> A1: Thanks for raising this concern which helps us to clarify the experimental settings. We guess this concern is caused by the misunderstanding on the "commonly used leave-one-out (LOO) setting." Here, the LOO setting in our work is a protocol for dataset split, which has no influence on input length during inference. Instead of using 8 frames as observations input for the model, we reduce them to 2 frames in both the training and inference. Please refer to Lines 261-270 for more details.
>
> Additionally, our motivation "There are many real-world situations where the model lacks sufficient time to observe" indicates that the observation is inadequate during inference but is not related to the training process. We can collect adequate trajectory points in an offline manner in training. But merely 2 points can be observed in inference in our setting.
>
> Overall, the experiment setting is consistent with the motivation.
>
> Besides, please kindly note that the setting is the same as that of MOE [2], the first work in instantaneous trajectory prediction.
>
> >Q2: Another concern is only human trajectories are predicted.
>
> A2: Thanks for your advice. Pedestrian trajectory prediction holds significance in autonomous driving [4], and the past decade has seen a surge in research efforts [5]. Our primary focus here is on pedestrian trajectory prediction. Diverging from prevailing approaches, we address a more challenging scenario, where only 2 frames are observed to predict future trajectories.
>
> We're now actively extending our method to predict trajectories of other moving agents like vehicles.
>
> >Q3: To better demonstrate the motivation, motivated experiments could be included, i.e., existing methods fail due to insufficient observations.
>
> A3: Following your suggestion, we conducted experiments using Trajectron++ and SGCN for better motivation. The results are listed in Table 1 of the PDF in Global Response. The Standard employs 8 frames to predict the next 12 trajectory points, while Instantaneous employs just 2 frames. Results reveal major drops when observed points decrease (e.g., 0.65 to 0.91 for SGCN on ETH/UCY). This accentuates the need for instantaneous trajectory prediction methods in autonomous driving. Remarkably, with only 2 frames as input, our BCDiff significantly enhances performance (e.g., 0.91 to 0.72 for SGCN on ETH/UCY).
>
> >Q4: It suggests that predicting historical and future paths can mutually benefit. This resembles [1], but lacks experiments, like separate predictions or sequential forecasting.
>
> A4: Thanks for your comments. We've incorporated discussions and citations in our latest revision. Our work diverges from [1] in terms of tasks and technical contributions. Our focus lies in addressing instantaneous trajectory prediction (with only 2 observed frames), whereas [1] deals with predicting future trajectories amidst occlusion (observation points are missing). In terms of technical contributions, we design two coupled diffusion models and propose a stepwise mutual guidance mechanism for generating both unobserved and future trajectories. In contrast, [1] employs a unified GCN model for predicting both missing and future trajectories.
>
> Actually, we have verified the efficacy of joint prediction in Table 3 of the submission. BCDiff-w/o.Guidance means two diffusion models for separate predictions. BCDiff-Bi.Guidance integrates the proposed bidirectional guidance mechanism for joint prediction. BCDiff-Bi.Guidance is better than BCDiff-w/o.Guidance, showing the efficacy of joint prediction.
>
> Following your suggestion, we add an experiment to sequentially forecast unobserved historical trajectories and future trajectories, denoted as BCDiff-two-step in Table 2 of the PDF in Global Response. BCDiff-Bi.Guidance is better than BCDiff-two-step, further indicating the superiority of joint prediction.
>
> > Q5: Why use LSTM in Eq.5, instead of the Bi-directional LSTM?
>
> A5: Sorry for confusing you. The LSTM in Eq.5 is the bi-directional LSTM. We will update it in the final version.
>
> >Q6: Details are lacking regarding the gap between training and inference. Training uses future ground-truth, but inference should be future-blind. Evaluation for the trained model is also unspecified.
>
> A6: Thank you for pointing out this concern which has improved the readability of our paper. Our method utilizes future ground-truth trajectories solely in training but doesn't use them in inference.
> We guess the misunderstanding is caused by the diffusion model [3] we employed. In training, we engage in both forward and backward processes. The former ensures ground-truth supervision, while the latter focuses on trajectory prediction.
> In inference, only backward is used for predicting future trajectories. We first initialize predicted future trajectories using Gaussian noises and exploit the trained network to gradually denoise $M$ times to obtain the final predicted trajectories. Please kindly refer to Algorithms 1 and 2 in [3] for the gaps between the training and inference in detail. We also include our training and inference details to clarify the gaps between them in Section C in the Appendix.
>
> To evaluate it, we use the trained network to predict future trajectories based on 2 observed points. Then, we calculate ADE & FDE between predicted and ground-truth future trajectories as the performance of the model.
>
> >Q7: I didn't see limitations in the submission.
>
> A7: We put the limitations in Section B of Appendix.
>
> Reference
>
> [1] Uncovering the Missing Pattern: Unified Framework Towards Trajectory Imputation and Prediction. CVPR'23
>
> [2] Human Trajectory Prediction with Momentary Observation. CVPR'22
>
> [3] Denoising Diffusion Probabilistic Models. NeurIPS'20
>
> [4] Codes for View Vertically: A Hierarchical Network for Trajectory Prediction via Fourier Spectrums. ECCV'22
>
> [5] Human Motion Trajectory Prediction: A Survey. ICRR'20

---

> > ### Comment · Reviewer_GWuJ · 2023-08-18
> > **Reply**
> >
> > Thank you for your comprehensive explanations. I checked all other reviewers' comments and your rebuttals. I think most of my concerns are addressed. Thus I will change my mind to lean toward the acceptance of this paper.
> >
> > I am not sure if you can update the draft as in previous years. But I expect you will revise accordingly and add what you showed in the rebuttals to the final version.

---

### Author Rebuttal · Authors · 2023-08-09

We really thank the reviewers for their valuable feedback. We are encouraged as reviewers found that our idea is novel and interesting (ytbj), promising (VvSx), solid (WQWs), and intuitive (JXan), our experiments show good results (GWuJ, ytbj, JXan) and are thorough (JXan), our paper is easy to follow (ytbj), easy to understand (WQWs), well-organized (GWuJ, VvSx, WQWs) and well-written (GWuJ, JXan). We have made the point-to-point response to each reviewer, and we include additional experimental results in the attached \textbf{PDF file} in the Global Response to assist reviewers in better understanding our responses.

Here, we list the additional experiments according to each reviewer's comments.

**[GWuJ, Q3]**. We include an experiment to better motivate our method. Results are shown in Table 1 in the attached PDF.

**[GWuJ, Q4]**. We include an experiment to further demonstrate the superiority of joint prediction. Results are shown in Table 2 in the attached PDF.

**[VvSx, Q1]**. We conduct an experiment to compare with the methods using the recently proposed SocialVAE as the backbone. Results are shown in the corresponding rebuttal thread.

**[WQWs, Q3]**. We provide some failure cases for instantaneous trajectory prediction, as shown in  Figure 2 in the attached PDF. Results are shown in the corresponding rebuttal thread.

**[JXan, Q1]**. We provide the performance of baseline that also utilized unobserved trajectories as supervision. Results are shown in the corresponding rebuttal thread.

**[JXan, Q3]**. We provide some hard cases for instantaneous trajectory prediction, as shown in Figure 1 in the attached PDF.

**[JXan, Q4]**. We provide the performance of unobserved trajectories when $T_{unobs}$ increases. Results are shown in the corresponding rebuttal thread.


Finally, we once again thank all reviewers for their insightful comments which are very helpful for improving the quality of our paper.

---

### Decision · Program_Chairs · 2023-09-21

**Decision:**

Accept (poster)

**Comment:**

The paper attempts to address a relevant problem space, but there are mixed feelings about the approach and its relative novelty. The proposed method revolves around completing the incomplete data and forecasting the future using diffusion learning. While the technique has been explored previously with marginal distinctions, the following points merit consideration:

Lack of Comprehensive Comparisons: The paper falls short in comparing its diffusion model with other contemporaneous models, particularly those presented at CVPR in recent years. This omission could detract from the paper's value, as the readers are left without a full understanding of how this model stacks up against the latest in the field.

Outdated Benchmarking: The benchmark models (Trajectron++, PCCSNet, and SGCN) selected for comparison in Tab1 are from 2020 and 2021. For a more robust assessment, the paper would have benefited from incorporating more recent encoder models and demonstrating that their method could achieve state-of-the-art results.

Positive Rebuttal Feedback: Notwithstanding the initial concerns, it's evident that the authors were responsive in their rebuttal. Their clarifications and additional inputs during the rebuttal phase have improved the perception of their submission, leading to an increase in reviewer ratings.

Given the constructive engagement of the authors during the rebuttal phase and the potential merit of their work, it is suggested that the paper be accepted . However, the authors are encouraged to take into account the feedback for future works and possibly provide more exhaustive comparisons with recent models to solidify their contributions.